# A systematic genome-wide mapping of oncogenic mutation selection during CRISPR-Cas9 genome editing

Sanju Sinha[1,2,3,9], Karina Barbosa [4,9], Kuoyuan Cheng[1,3,9], Mark D. M. Leiserson[3], Prashant Jain[4], Anagha Deshpande[4], David M. Wilson III[5], Bríd M. Ryan [2], Ji Luo [6], Ze'ev A. Ronai[4], Joo Sang Lee [7,8,10], Aniruddha J. Deshpande [4,10 ✉] & Eytan Ruppin[1,10 ✉]

Recent studies have reported that genome editing by CRISPR–Cas9 induces a DNA damage response mediated by *p53* in primary cells hampering their growth. This could lead to a selection of cells with pre-existing *p53* mutations. In this study, employing an integrated computational and experimental framework, we systematically investigated the possibility of selection of additional cancer driver mutations during CRISPR-Cas9 gene editing. We first confirm the previous findings of the selection for pre-existing *p53* mutations by CRISPR-Cas9. We next demonstrate that similar to *p53*, wildtype *KRAS* may also hamper the growth of Cas9-edited cells, potentially conferring a selective advantage to pre-existing *KRAS*-mutant cells. These selective effects are widespread, extending across cell-types and methods of CRISPR-Cas9 delivery and the strength of selection depends on the sgRNA sequence and the gene being edited. The selection for pre-existing *p53* or *KRAS* mutations may confound CRISPR-Cas9 screens in cancer cells and more importantly, calls for monitoring patients undergoing CRISPR-Cas9-based editing for clinical therapeutics for pre-existing *p53* and *KRAS* mutations.

[1] Cancer Data Science Laboratory, Center for Cancer Research, National Cancer Institute, National Institutes of Health, Bethesda, MD 20892, USA. [2] Laboratory of Human Carcinogenesis, Center for Cancer Research, National Cancer Institute, National Institutes of Health, Bethesda, MD 20850, USA. [3] Center for Bioinformatics and Computational Biology, University of Maryland, College Park, MD 20742, USA. [4] Tumor Initiation Program, Cancer Center, Sanford Burnham Prebys Medical Discovery Institute, La Jolla, CA 92037, USA. [5] Laboratory of Molecular Gerontology, National Institute on Aging, Intramural Research Program, National Institutes of Health, Baltimore, MD 21224, USA. [6] Laboratory of Cancer Biology and Genetics, Center for Cancer Research, National Cancer Institute, National Institute of Health, Bethesda, MD 20850, USA. [7] Samsung Medical Center, Sungkyunkwan University School of Medicine, Suwon 16419, Republic of Korea. [8] Department of Precision Medicine, School of Medicine and Department of Artificial Intelligence, Sungkyunkwan University, Suwon 16419, Republic of Korea. [9] These authors contributed equally: Sanju Sinha, Karina Barbosa, Kuoyuan Cheng. [10] These authors jointly supervised this work: Joo Sang Lee, Aniruddha J. Deshpande, Eytan Ruppin. ✉email: adeshpande@sbpdiscovery.org; eytan.ruppin@nih.gov

Clustered regularly interspaced short palindromic repeats (CRISPR) enables targeted gene disruption and editing, a powerful technology that expands our understanding of fundamental biological processes[1]. Beyond its impact on biological research, CRISPR-based approaches have been considered for various applications in medicine, from reparative editing of primary cells to the development of new strategies to treat a variety of genetic diseases, including cancer. However, several clinical trials based on CRISPR technology have been deferred due to significant potential risks, including off-target effects[2-4], generation of unexpected chromosomal alterations[5], and potential immunogenicity[6]. Other studies have demonstrated that double-stranded breaks (DSBs) induced during CRISPR-Cas9-based gene knockout (CRISPR-KO) can lead to DNA damage response, whose level can either be associated with the copy number of the targeted gene[7-10] or in some cases structural rearrangements in the region[11,12].

Recent studies have shown that the DNA damage response following CRISPR-KO can be mediated by *p53*, a tumor-suppressor gene mutated in over 50% of all human cancers[13,14]. Genome-wide CRISPR screening in immortalized human retinal pigment epithelial (RPE1) cells[14] revealed that a *p53*-mediated DNA damage response, followed by cell cycle arrest, is induced upon generation of DSBs by the Cas9 endonuclease, favoring the survival of cells that have inactivated the *p53* pathway. Most recently, a study showed that exogenous expression of Cas9 can also activate this *p53*-mediated DNA damage response[15]. While these studies indicate that CRISPR-Cas9 genome editing techniques may select for *p53*-mutated cells[13-15], several outstanding questions remain unaddressed: First, since most of these *p53* studies have involved only a small number of primary or transformed cells[13,14], it is unclear whether *p53* selection can happen broadly across multiple different cell types including transformed cancer cells. Second, it is not clear whether stronger *p53* selection can happen when certain genes or parts of the genome are targeted, or the level of selection is more homogenous regardless of the genes being edited. And finally, it remains to be investigated whether this selection is limited to *p53* only or that other cancer driver genes can also be selected for during CRISPR-Cas9 genome editing.

To address these questions, here we employ a computational framework coupled with experimental validations to conduct a comprehensive evaluation of each cancer driver mutation-selection associated with CRISPR-Cas9. We first demonstrate that CRISPR-KO-induced mutant *p53* selection can be observed in transformed and non-transformed cells of diverse lineages via both lentivirus and ribonucleoprotein (RNP)-based Cas9 delivery. More importantly, we systematically characterized mutation-selection in other cancer driver genes during CRISPR-Cas9 identifying that *KRAS* mutants can also be selected for, as demonstrated in large-scale genetic screens and Cas9-expressing cell lines. We further identified the underlying pathways that are likely to mediate this selection.

## Results

### CRISPR-Cas9 gene-knockouts selects for p53 mutations in a vast variety of transformed and non-transformed cell types. We first sought to address two important gaps in our understanding of CRISPR-KO-driven mutant *p53* selection—firstly, we wanted to investigate whether this selection generalizes across cell types. Secondly, we wanted to understand what type of sgRNAs, genes, and gene networks drive this selection. We analyzed the DepMap[16] genome-wide gene essentiality data across 248 cancer cell lines (Supplementary Data 1), where both CRISPR-Cas9 (*AVANA*[10]) and shRNA-based (*Achilles*[16]) genetic screens were conducted. We searched for genes whose CRISPR-Cas9-

based knockout (CRISPR-KO) reduced cell viability more (i.e. more essential) in *p53*-wildtype (WT; $N = 75$) than *p53*-mutant ($N = 173$) cell lines, but do not exhibit such differential essentiality in the shRNA-based screens (see the "Methods" section). The KO of such genes may select for *p53* mutants specifically during CRISPR-Cas9 editing. In the CRISPR-Cas9 screen, we find many more genes (981) that are more essential in *p53*-WT vs. *p53*-mutant cell lines, compared to the genes that are more essential in *p53*-mutant cells (237 genes). In contrast, the numbers of such differentially essential genes in the shRNA screens were balanced (~1500 each). Such significantly different patterns between CRISPR-Cas9 and shRNA screens (Fig. 1a left panel, Chi-squared test $P < 1.4E-284$) points to a bias that knockout/knockdown of a gene is more likely to impair the fitness of *p53*-WT cells specifically with CRISPR-Cas9 but not with shRNA.

Among the 981 genes that are more essential in *p53*-WT cells with CRISPR-KO, 861 genes (87%) do not exhibit this differential essentiality in shRNA screens. We hence termed these *CRISPR-specific differentially essential positive (CDE+)* genes (Fig. 1a right panel; genes listed in Supplementary Data 2A). We find that these CDE+ genes are preferentially located in chromosomal bands containing common fragile sites (CFSs; hypergeometric $P < 2.3E-4$, Fig. 1b, Supplementary Data 3), which are prone to replicative stress, fork collapse and DNA breaks that cause genomic instability (GI)[17]. As CRISPR-KO could induce kilobase-scale structural alterations near the targeted site[18], this finding suggests that CRISPR-targeting near CFSs may enhance DNA damage, promote the *p53*-dependent cell death response and provide a selective advantage to *p53*-mutant cells. The sgRNAs of the CDE+ genes also tend to target highly accessible chromatin (HAC) (hypergeometric $P < 0.02$; see the "Methods" section), thus inducing a strong damage response. The top pathways enriched within CDE+ genes include DNA damage response, DNA repair, and Fanconi anemia (FA; hypergeometric test adjusted $P < 0.01$, Supplementary Data 2B). This is consistent with the recent report that the FA pathway is involved in repairing Cas9-induced DNA double-strand breaks (DSBs)[19] and that their KO may further enhance DNA damage.

Analogous to CDE+, we defined CDE− genes, which are more essential in *p53*-mutant (vs. WT) cells with CRISPR-KO, but do not show such difference in shRNA screens (185 genes, right panel of Fig. 1a). CDE− genes are involved in cellular processes that engage *p53*, including mitotic checkpoints, DNA replication and cell cycle (Supplementary Data 2B, Fig. 1d, hypergeometric test adjusted $P < 0.1$), with the top hit being the key cell cycle regulator *CDKN1A*[13,14] (a.k.a. *p21*, Wilcoxon rank-sum $P < 1.85E-08$, Fig. 1c). Transiently inhibiting CDE− genes during CRISPR-KO may mitigate *p53* mutation selection and could be of interest from a translational point of view. Top CDE+/− genes are highlighted in Fig. 1c.

We next tested for a series of confounding factors that can potentially lead to this skewness (expanded details in Supplementary Notes 1, 2, 6). We repeated our analysis in the following modes to control for: 1. Gene copy number variations in cell lines, by correcting for gene copy number via a linear regression while testing whether a gene is differentially essential in WT vs. mutant cell lines (Supplementary Fig. 1), 2. Functional effect of different *p53* variants, by focusing on cell lines harboring known *loss-of-function* (LOF) *p53* mutations solely (see the "Methods" section, $N = 78$) vs. WT (Supplementary Note 1a), 3. Effect of partial vs. complete silencing of gene expression in case shRNA-KD and CRISPR-Cas9 KO, respectively, by using genes that are not expressed at all (Supplementary Note 1b), 4. Effect of a potential functional relationship, e.g. synthetic lethal or rescue interaction with p53, by using non-essential genes (Supplementary Note 1c), 5. Effect of different reagents and sgRNA depletion time, by using

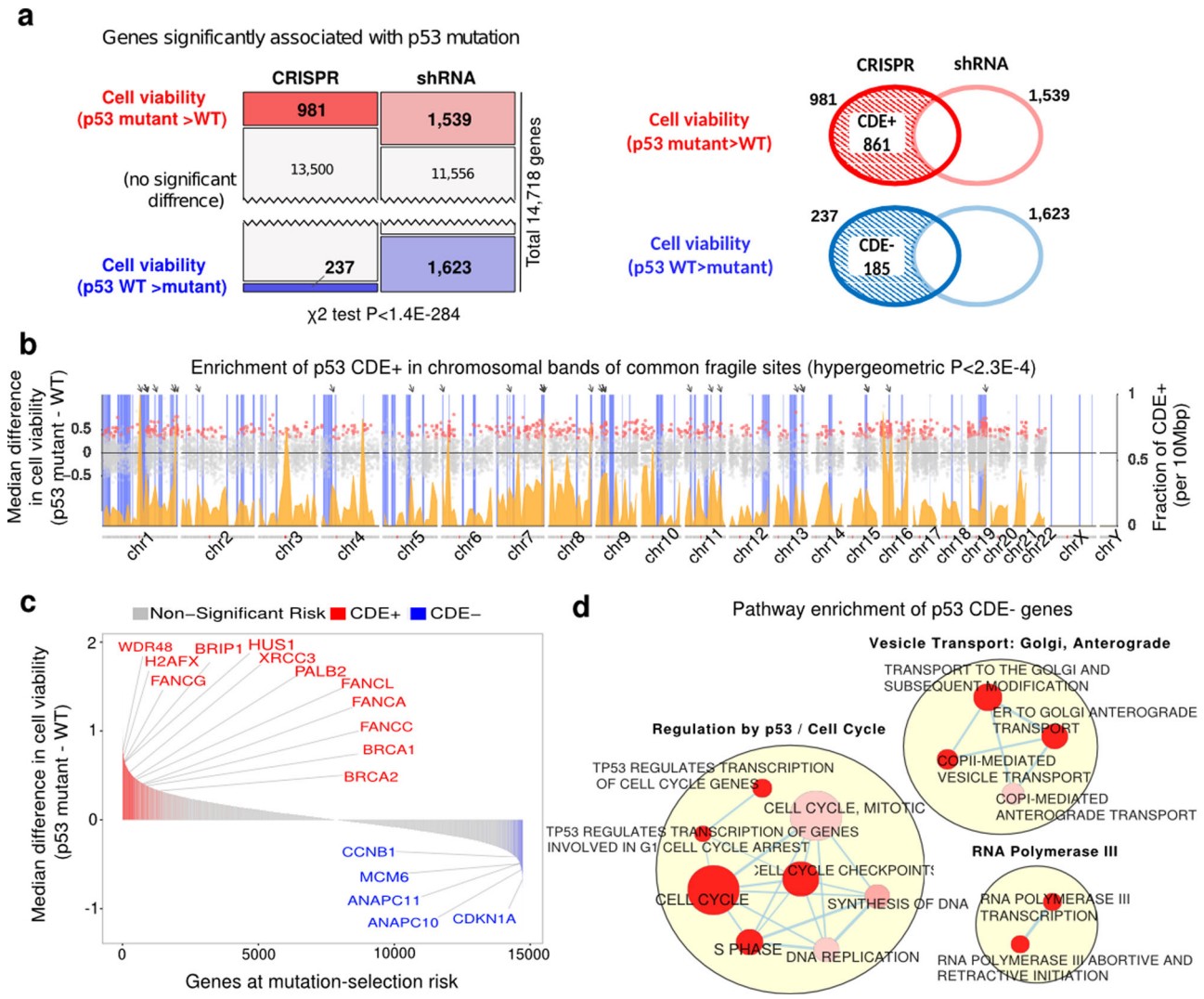

**Fig. 1 A genome-wide view of *p53*-mutant selection. a** Upper panel: number of genes whose essentiality is significantly associated with *p53* mutation status in CRISPR and shRNA screens (one-sided Wilcoxon rank-sum has been performed with FDR threshold of 0.1). Lower panel: the definition of CDE+ and CDE− genes. **b** Enrichment of *p53* CDE+ genes in common fragile sites (CFSs). The *x*-axis denotes the chromosomal position; the scatter plot (*y*-axis on the left-hand side) shows the difference of median post-CRISPR-KO cell viability values in *p53* mutant vs. *p53* WT cell lines for *p53* CDE+ genes (red dots) and all other genes (gray dots); the density plot (colored orange, *y*-axis on the right-hand side) shows the fraction of *p53* CDE+ genes among all genes per DNA segments of 10 Mbp along the genome; the vertical blue bars indicate the chromosomal bands of CFSs, and prominent sites where peaks of high CDE+ gene density coincide with CFSs are marked by arrows on the top. **c** The distribution of predicted level of CRISPR-Cas9 *p53*-mutant selection across the genome. Significant CDE+ genes that are part of the FA pathway are marked in red and significant CDE− genes that are part of cell cycle regulation are in blue. **d** Visualization of the pathways enriched for *p53* CDE− genes where significance is calculated using the GSEA method as implemented in the R package fgsea[38]. Only significantly enriched pathways (FDR < 0.1) specific to CRISPR (and not in genes showing differential essentiality in the shRNA screens are shown). Pathways are depicted as nodes whose sizes correlate with pathway lengths and colors represent enrichment significance (the darker, the more significant). Pathway nodes are connected and clustered based on their functional similarities, and higher-level functional terms are given for each of the clusters (see the "Methods" section). For clarity, only the largest clusters are shown.

independent CRISPR-Cas9 screen (Sanger Screen) performed using 326 cancer cell lines[20] (Supplementary Note 2, Supplementary Fig. 2), 6. Off target effect of sgRNA, by computing and taking into account an off-target score of the sgRNA used in the screens (Supplementary Note 6). We showed that our findings via the CDE identification process remain valid after controlling for this broad array of potentially confounding factors.

We next performed our own CRISPR-Cas9 essentiality screen, employing CRISPRi-based essentiality screens as a control in a pair of *p53*-isogenic MOLM13 leukemia cell lines (WT and *p53* R248Q mutant). We used a deep (10 guides per gene) and focused sgRNA library targeting top *p53* CDE+ and CDE− genes (see the "Methods" section; Fig. 2a, details in Supplementary Note 10,

Supplementary Data 4). Here, we observed in the CRISPR-Cas9 screen that the CDE+ genes are more essential in *p53*-WT vs. mutant cells, and vice versa for the CDE− genes (Wilcoxon signed-rank test $P < 0.08$ and $P = 0.03$ for CDE+/− genes respectively). Reassuringly, we do not see such differential essentiality in the CRISPRi screens (Wilcoxon signed-rank $P = 0.32$ and $0.29$; Top 10% CDE+/− genes are depicted in Fig. 2b).

To further assess whether such selection effects can be observed in non-transformed cells, we next tested and observed that indeed our *p53* CDE+ genes have higher essentiality in WT vs. isogenic-mutant cells in published CRISPR-Cas9[12] but not shRNA[47] genome-wide screens performed in non-transformed RPE1 cells

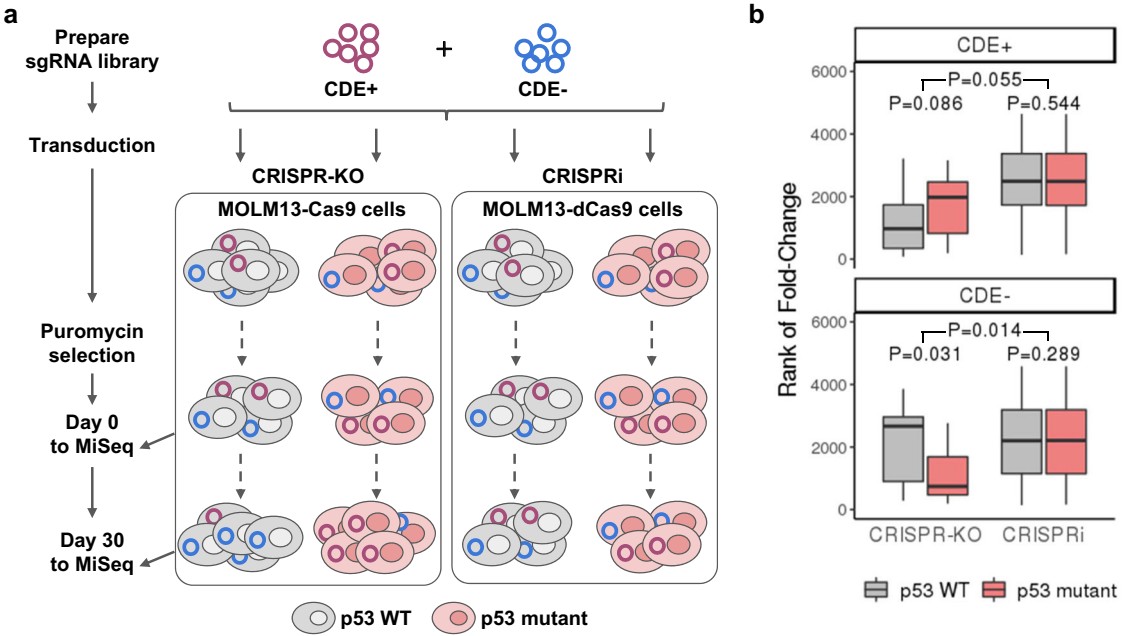

**Fig. 2 Validation of *p53* CDE genes in isogenic MOLM13 cell lines via pooled CRISPR screens. a** A flowchart showing the experimental procedure of CRISPR-KO and CRISPRi screening of pooled *p53* CDE+/− genes in a pair of *p53*-isogenic MOLM13 cell lines (see the "Methods" section for details). **b** The day 30 to day 0 fold-change (converted to rank) of reads corresponding to the sgRNAs for *p53* CDE+ genes (upper panel) and CDE− genes (lower panel), in *p53* WT MOLM13 cells (gray boxes) vs. the isogenic *p53*-mutant cells (red boxes) for the CRISPR-KO and CRISPRi screenings, respectively. 10 guides per gene were designed for the top 200 of each CDE+ and CDE− genes and were used to derive statistics here. The bottom *P* values are for two-sided Wilcoxon signed-rank tests comparing *p53* WT and mutant cells, the upper ones are *P* values of non-parametric tests comparing the difference of *p53* mutant and WT rank values between CRISPR-KO and CRISPRi experiments. In the boxplots, the center line, box edges, and whiskers denote the median, interquartile range, and the rest of the distribution in respective order, except for points that were determined to be outliers using a method that is a function of the interquartile range, as in standard box plots.

(Supplementary Fig. 3, screens quality control discussed in Supplementary Note 9 and Supplementary Fig. 4). This finding is further confirmed by mining seven CRISPR-KO genome-wide screens[21], including two *p53*-null and five *p53*-WT RPE1 cells screens (Supplementary Fig. 5, Supplementary Note 3).

**A competition assay shows selection for p53 mutant over wildtype cells following CRISPR-Cas9 knockout of CDE+ genes.** To test whether the CRISPR-KO of CDE+ genes leads to the selection of *p53*-mutant cells in a competitive setting[22–24], we silenced the top five predicted CDE+ genes using CRISPR-Cas9 and CRISPRi in the *p53*-isogenic MOLM13 cells (see the "Methods" section). Following a lentiviral sgRNA transduction, the WT and mutant cells were mixed at an initial ratio of 95:5, and monitored by flow cytometry for up to 25 days (illustrated in Fig. 3a; Supplementary Data 5A). Silencing 2/5 CDE+ genes (*NDUFB6* and *NDUFB10*) induced a strong progressive *p53* mutant enrichment of up to five folds over WT at day 25 specifically in CRISPR-KO, across several independent sgRNAs and not for NTC (Fig. 3b, blue lines). No inverse enrichment in *p53* WT cells was observed in the competitive assays involving the three other CDE+ genes (Supplementary Fig. 6). We observed that sgRNAs targeting *NDUFB6* induced significantly higher DNA damage compared to NTC-treated cells specifically in p53 WT cells (Supplementary Fig. 7, despite higher editing efficiency in the mutant cells as shown in Fig. 4c), demonstrating that the DNA damage was not just due to Cas9 expression. This may partly explain their selective competitive advantage upon the CDE+ gene KO. Testing the robustness of this competitive selection advantage for *p53*-mutant cells, we repeated the CRISPR-KO competitive assay for a larger number of 18 top CDE+ genes with up to four unique sgRNAs per gene

and monitored the assay up to 15 days (Supplementary Data 5B). Using the non-targeting sgRNA as a baseline, we observed the competitive outgrowth of *p53*-mutant cells for 15 out of 28 sgRNAs and 10 out of 18 CDE+ genes tested (Fig. 3c).

**p53 mutation-selection phenomena extend to transient knockout and primary cells.** We next asked whether our top CDE+ genes may also select for p53 mutants under CRISPR-Cas9 transient knockout. We delivered Cas9 and the sgRNA as an RNP. We observed that upon Cas9-RNP mediated transfection of a sgRNA targeting our top CDE+ gene from our pooled and competition assays, *NDUFB6* (see the "Methods" section), there was a higher loss of edited cells in the *p53* WT vs isogenic *p53*-mutant MOLM13 cells over 10 days of culture, as measured by the change in ICE scores (Fig. 4a top panel). Using an orthogonal method of proliferation monitoring by dye-dilution[25], we observed that there was a progressive slowing down in cell proliferation of *p53* WT, but not *p53* mutant MOLM13 cells upon Cas9-RNP based KO of *NDUFB6* vs. respective non-targeting controls (NTCs) (Fig. 4b, c). Similar to the lentiviral system, this is likely due to the DNA damage induced by the *NDUFB6* sgRNA compared to the Cas9 only or Cas9 with NTC controls in *p53* WT cells (Figs. 4b and S8). We repeated this transient knockout in non-transformed cells (RPE1) and consistently observed an increased loss of edited *p53* WT over *p53*-mutant cells (Fig. 4a, bottom panel). Notably, we also observed a selection of *p53* mutant over WT in patient tumors profiles (TCGA) based on the copy number alteration patterns of CDE+ genes (details in Supplementary Note 11A).

**KRAS mutant cell lines exhibit a selection advantage in large-scale genetic screens.** To determine whether additional cancer

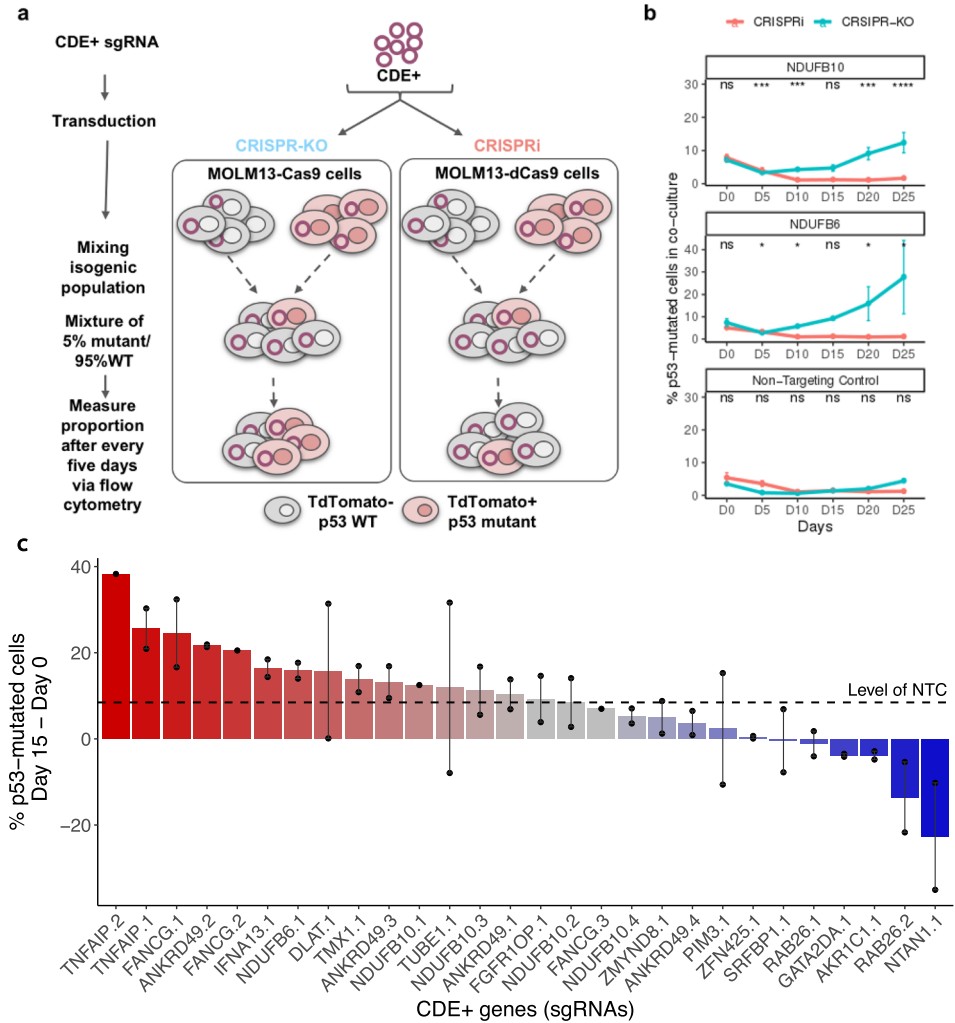

**Fig. 3 Selection for *p53* mutant cells under CRISPR-Cas9 knockout of CDE+ genes in a co-culture of *p53* WT/mutant cells. a** An illustration showing the experimental design of the competition assay where isogenic *p53* WT/mutant MOLM13 cell lines were mixed with a ratio of 95:5 and top *p53* CDE+ genes were knocked out by CRISPR-Cas9. The population ratio was monitored for 25 days at a 5-day interval starting from the day of sgRNA transduction. **b** Change in the percentage of *p53* mutant cells in the p53-mutant WT cells (y-axis) co-culture with time (x-axis, number of days in co-culture), under the CRISPR-KO or CRISPRi of individual selected top p53 CDE+ genes or with non-targeting control sgRNA. The p-values are calculated using two-sided Wilcoxon Rank Sum tests. Data are presented as mean values ± SEM across N = 2 independent biological replicates (unique sgRNAs). **c** The difference of the percentage of *p53*-mutant cells between Day 15 and Day 0 in co-culture (y-axis), under the CRISPR-KO of a larger set of top *p53* CDE+ genes (x-axis, by individual sgRNAs, specified by number suffixes after gene symbols). Similar to panel (**a**), data are presented as mean values where the error bars represent standard error across replicates for each sgRNA. Data are presented as mean values ± SEM across N = 3 independent biological replicates (unique sgRNAs). The horizontal dashed line represents the value for non-targeting control sgRNA (NTC).

driver mutations may be selected for the following CRISPR-KO, we focused on a list of 61 cancer driver genes from Vogelstein et al.[26] that are mutated in at least 10 of the cell lines screened in the AVANA[10] and Achilles[16] datasets. For each of these cancer genes, we identified the differentially essential genes between its WT and mutant cell lines in the CRISPR-Cas9 (AVANA) and shRNA (Achilles) screens, as described above for p53. We ranked the cancer genes by the significance of skewness in the numbers of differentially essential genes from CRISPR-Cas9 vs. shRNA screens similar to that shown in Fig. 1a for *p53* (with Fisher's exact tests, see the "Methods" section; results shown in Fig. 5a and Supplementary Data 6). The mutants of these genes may be selected for during CRISPR-KO, as their WT cells are overall more vulnerable during CRISPR-KO compared to the mutants. We term these genes "(potential) CRISPR-selected cancer drivers" (CCDs). The top significant CCD in addition to *p53* is the oncogene *KRAS*. Like for *p53*, potential confounding factors

including copy number were controlled for (Supplementary Note 1, Supplementary Fig. 1b), and there is no significant correlation between the mutation profiles of *KRAS* and *p53* (Fisher's test $P = 0.67$), suggesting that *KRAS* might be a CCD independent of *p53*. We thus next focused on investigating the selection of mutant *KRAS* as another major CCD.

*KRAS* is a major oncogene whose gain of function mutation is known to activate various DNA repair pathways and may override the trigger of cell death upon DNA damage[27,28], supporting its role as a CCD. We computationally identified the CDE+ and CDE− genes of *KRAS* in a similar way described above for *p53* (Fig. 5b, Supplementary Data 2A). *KRAS* has high numbers of CDE+/− genes, while only very few *KRAS* mutation-associated genes are identified in the shRNA screen. The predicted median mutant selection levels are comparable to those of *p53* (Supplementary Note 4), i.e. the CRISPR-KO of its CDE+ genes is likely to drive comparable levels of mutant

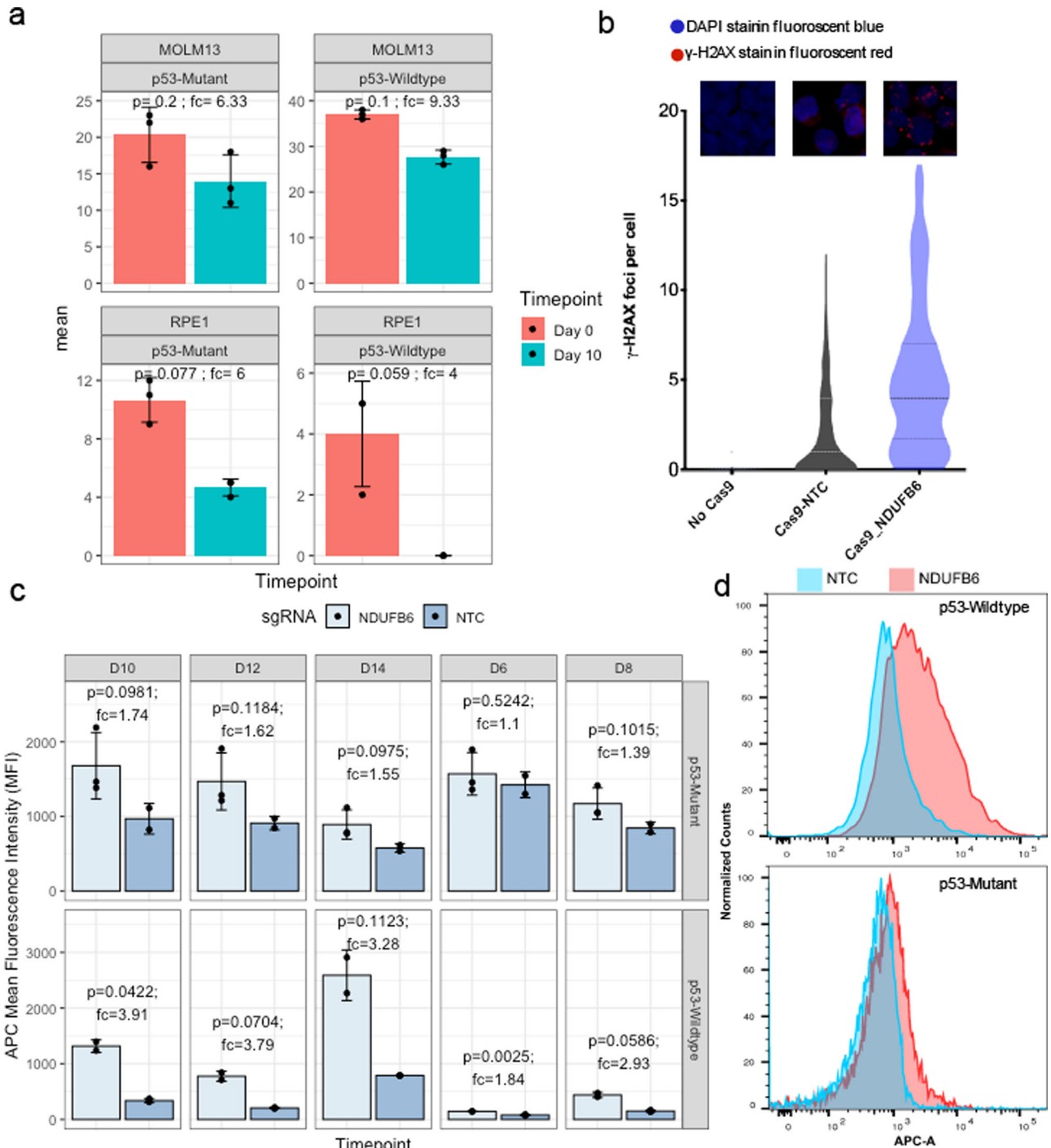

**Fig. 4 Transient knockout in both MOLM13 and RPE1 leads to preferential loss of p53 WT over mutant cells. a** The editing efficiency of NDUFB6 around the sgRNA cut site was determined in *p53* mutant and wildtype isogenic pair of MOLM13 (top panels, transformed cells) and RPE1 cells (bottom panels, primary cells) using ICE protocol (see the "Methods" section) at day 0 (orange) and day 10 (green) after Cas9-RNP-sgRNA nucleofection. Differences in day 0 compared to day 10 editing efficiency can be used as a measure of relative fitness of edited compared to non-edited cells. The *p*-values are calculated using two-sided Wilcoxon Rank Sum tests. Data are presented as mean values ± SEM across N = 4 independent biological replicates (unique sgRNAs). **b** DNA damage is quantified from gamma H2AX (γ-H2AX) staining images and measured by gH2AX staining in *p53* wildtype MOLM13 cells with no Cas9, Cas9 + sgRNA for a non-targeting control (NTC) or NDUFB6. γ-H2AX foci (*y*-axis) in all three conditions (*x*-axis) are enumerated in the violin plot. **c** Mean fluorescence intensity of the CellTrace™ dye (APC) in MOLM13 p53 mutant (top panel) vs. wildtype cells (bottom panel) is shown for NTC (light blue) or *NDUFB6* targeting sgRNAs (dark blue). CellTrace™ APC fluorescence is inversely correlated with proliferation. The error bars denote standard error (mean ± standard deviation) across three replicates. The p-values are calculated using a two-sided *t*-test given a small number of data points (*n* = 3). **d** Proliferative effects of *NDUFB6* editing in RNP-transfected *p53* mutants compared to wild-type cells. A histogram of MOLM13 *p53* wildtype (top panel) cells transfected with an NTC or an *NDUFB6* sgRNA is shown with the fluorescence intensity of the CellTrace™ dye (APC) on the *x*-axis. Similarly, MOLM13 *p53* mutant cells are plotted in the bottom panel. The error bars are presented and computed as panel (**a**).

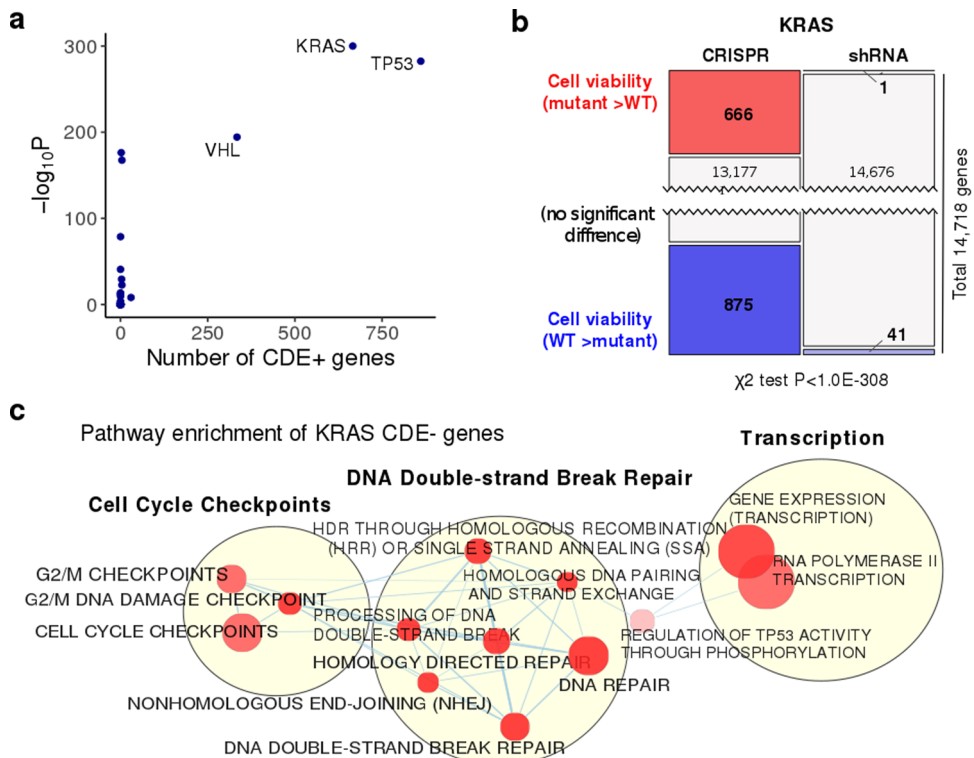

**Fig. 5 Large-scale genetic screening identifies KRAS as a second major cancer driver whose mutation can be potentially selected for by CRISPR-Cas9.**
**a** A scatter plot showing the number of identified CDE+ genes (x-axis) and the negative log10-transformed P values of one-sided Fisher's exact test (y-axis) testing for the imbalance in the number of differentially essential genes in CRISPR and shRNA screens for the 61 major cancer driver genes from Vogelstein et al. [27]. p53 and KRAS are identified as the top two significant cancer genes with a higher number of CDE+ genes. **b** The number of genes whose essentiality is significantly associated with KRAS mutational status in CRISPR and shRNA screens (P = 1.0E−208, one-sided Wilcoxon rank-sum has been performed with FDR threshold of 0.1). **c** Visualization of pathways enriched for KRAS CDE-genes where significance is calculated using the GSEA method as implemented in the R package fgsea21 . Only significant pathways (FDR > 0.1) specific to CDE and not to the genes showing differential essentiality in the shRNA screens are included. Pathways are shown as nodes whose sizes correlate with pathway lengths and colors represent the significance of their enrichment (the darker the more significant). Pathway nodes are connected and clustered based on their functional similarities, and higher-level functional terms are given for each of the clusters (Methods). For clarity, only the largest clusters are shown.

selection as the KO of the CDE+ genes of p53. Fourteen genes are CDE+ genes of both p53 and KRAS, and thus their CRISPR-KO may impose considerable selection for both KRAS and p53 mutants (Supplementary Data 2A). Consistent with the knowledge of downstream pathways regulated by activated KRAS[27,28], its CDE− genes are significantly enriched for DNA DSB repair pathways (FDR < 0.02, see the "Methods" section, visualized in Fig. 5c, Supplementary Data 2E).

Based on the number of CDE+ genes (Fig. 5a), the third-ranked CCD is VHL, having a large number of CDE+ genes. Like p53 and KRAS, we controlled for a series of potential confounding factors (Supplementary Notes 1, 2, 6). We also note that the CDE+ genes of VHL were also enriched in chromosomal bands of CFSs (hypergeometric P < 2.4e−2). The known function of VHL as a positive regulator of p53 in DNA damage-induced cell cycle arrest or apoptosis[13] possibly accounts for its role as a CCD. However, in this study, we chose to carefully study one additional CCD and our top-ranked hit—KRAS and its potential mutant selection during CRISPR-Cas9 gene editing and thus will focus on that from here onwards.

**Pooled essentiality screens show that KRAS CDE+ genes are more essential in WT than mutants and vice versa for CDE− genes specifically in CRISPR-Cas9 screens.** Similar to p53, we next performed our own CRISPR-Cas9 and a control CRISPRi gene essentiality screens, but on a smaller scale, in a pair of isogenic KRAS WT and KRAS G12D mutant MOLM13 cell lines

using a sgRNA library targeting top KRAS CDE+ and CDE− genes (details in Supplementary Note 10, Supplementary Data 7). Here, we observed that the KRAS CDE+ genes are more essential in WT than mutants and vice versa for CDE− genes specifically in CRISPR-Cas9 screens (Wilcoxon signed-rank test P < 0.074 and P < 0.042 for CDE+/−, respectively, Fig. 6a left panel), but not in CRISPRi screens (Wilcoxon signed-rank P < 0.22 and P < 0.49; Fig. 6a right panel). Similar results were obtained from analyzing published genome-wide CRISPR-Cas9[29] and shRNA genetic screens[30] performed in a different pair of KRAS isogenic cell lines (WT and G12D mutation in DLD1 cell line; Fig. 6b). Similar to p53, we also observed a selection of KRAS mutants in patient tumor profiles (TCGA[31]) based on the copy number alteration patterns of its CDE+ genes (details in Supplementary Note 11B).

**A competition assay shows selection for KRAS mutant over wildtype cells following CRISPR-Cas9 knockout of KRAS CDE + genes.** To test whether, like the p53 case, CRISPR-KO of KRAS CDE+ genes can confer a selective advantage to KRAS mutant over WT cells in co-culture, we conducted a similar competition assay using a pair of WT and KRAS G12D mutant isogenic MOLM13 cell lines. As in the experiment for p53, we mixed the WT and KRAS mutant cells at an initial ratio of 95:5 following KRAS CDE+ sgRNA transduction and monitored the population for 15 days to track the percentage of KRAS-mutant cells (TdTomato+) with flow cytometry (see the "Methods" section;

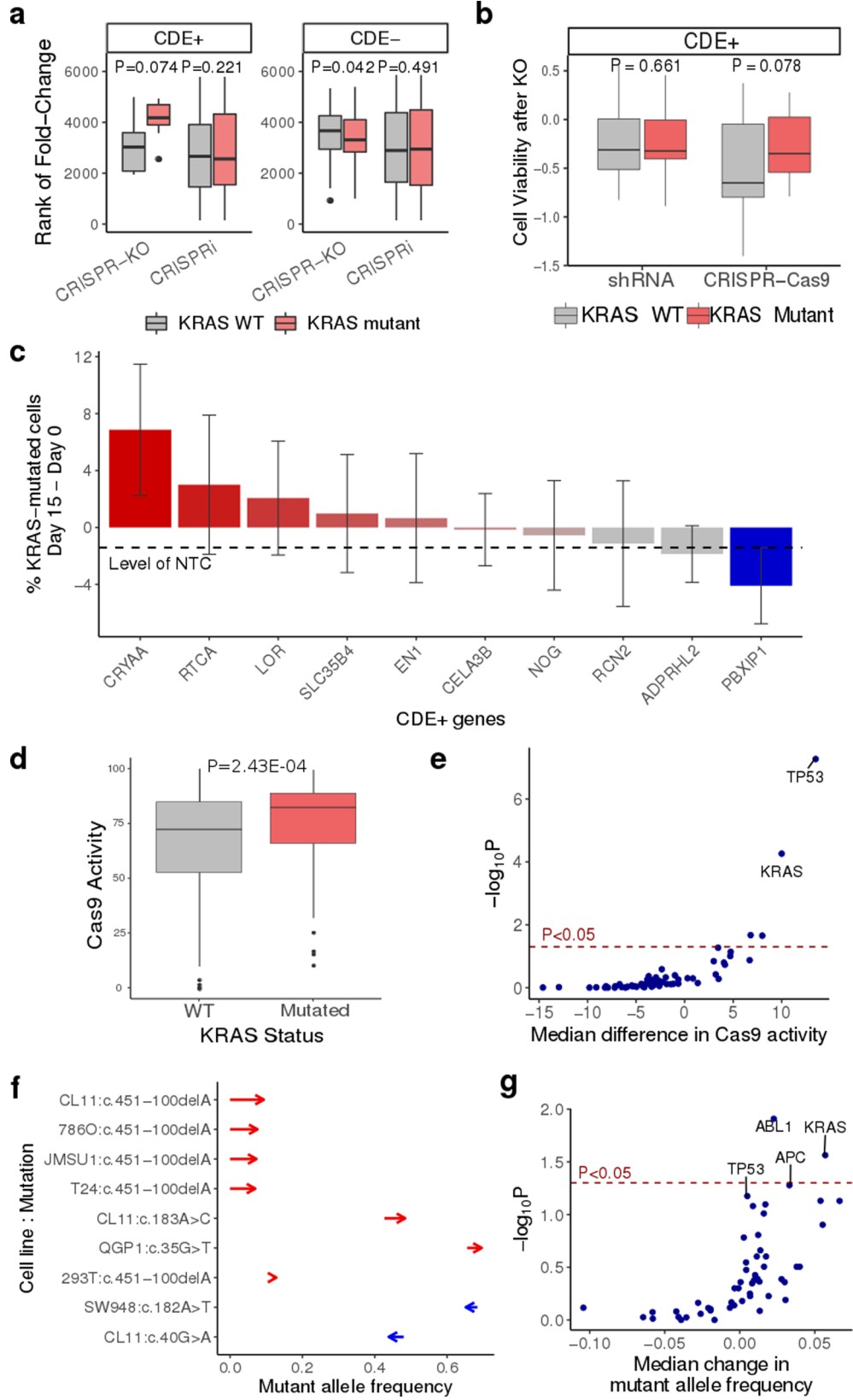

Supplementary Data 8). A total of 10 *KRAS* CDE+ genes were tested, in addition to NTC. In the control group, the *KRAS* mutant cell fraction decreased with time, indicating that the mutant cells have lower baseline fitness levels than the WT cells.

In comparison, in 8 out of 10 CDE+ genes tested, there is a gain in the fitness of the mutant cells (Fig. 6c; see the "Methods" section), testifying that even though the *KRAS*-mutant cells have a lower baseline fitness level, the CRISPR-KO of the majority of

**Fig. 6 Beyond p53: experimental evidence identifying KRAS as another major cancer driver whose mutation can be potentially selected for by CRISPR-Cas9. a** CRISPR-Cas9 and CRISPRi screens of the top KRAS CDE gene knockouts were performed in isogenic MOLM13 and MOLM13-KRAS-G12D cell lines. The box plot shows the trend that the sgRNAs of the KRAS CDE+ genes are more depleted in KRAS WT cells vs KRAS mutant cells and vice versa for KRAS CDE- genes in CRISPR-Cas9 screens, but there is no such trend in the CRISPRi screens. The P values shown are of one-tailed Wilcoxon signed-rank tests. Top 200 candidates of both CDE+ and CDE- genes were used to derive this statistic with $N = 10$ unique sgRNA per gene. **b** Analysis of published genome-wide CRISPR-Cas9 and shRNA screens in KRAS-isogenic DLD1 cell line. The box plot shows the trend that the CRISPR-KO of KRAS CDE+ genes reduces cell viability more in KRAS WT cells than KRAS mutant cells, while there is no such trend in the shRNA screen. The P values of one-tailed Wilcoxon signed-rank tests are shown. 861 CDE+ and 185 CDE-genes were used to derive this statistic with $N = 3$ unique sgRNA per gene. **c** The mean difference in the percentage of KRAS mutant cells between Day 15 and Day 0 in co-culture (y-axis), under the CRISPR-KO of different KRAS CDE+ genes (x-axis). The y-axis values were obtained by fitting a linear model for each gene, with the percentage of KRAS mutant cells as dependent variable and time (day, as a continuous variable) and sgRNA as independent variables. The linear model coefficients associated with the variable "day" multiplied by 15 are plotted. Error bars represent standard errors of the coefficients as estimated from the linear models. NTC: non-targeting control sgRNA. $N = 4$ unique sgRNAs were used per gene. NTC: non-targeting control sgRNA. **d** A box plot showing the comparison of stable Cas9 activity measured via GFP reporter assay10 in $N = 1375$ WT vs $N = 226$ KRAS (distribution of total data points) mutant cancer cell lines, two-sided Wilcoxon rank-sum test P-value 2.4E−04. **e** A scatter plot showing the difference in Cas9 activity between cell lines with a driver WT vs mutant for each driver (effect size, x-axis) and a corresponding Wilcoxon two-sided significance for this difference (negative log10-P-value, y-axis). **f** The change in mutant allele frequency (x-axis) of the KRAS mutations detected in different cell lines (cell line-mutation pair on the y-axis) after induced Cas9 expression, compared to the corresponding parental cell lines, based on data from Enache et al. [13]. The starts and ends of arrows correspond to the mutant allele frequencies in the parental and the Cas9-expressed cell lines, respectively. Cases of increased allele frequency are colored in red, and those with decreased frequency are colored in blue. **g** A scatter plot showing the median change in mutant allele frequency after induced Cas9 expression across all cell lines in [13] (x-axis) and the corresponding Wilcoxon signed-rank test significance (negative log10-P value, y-axis) for the 61 major cancer driver genes from Vogelstein et al. [27]. The p-values are calculated using two-sided Wilcoxon Rank Sum tests unless not specified otherwise. In the boxplots of panels **a, b,** and **d**, the center line, box edges, and whiskers denote the median, interquartile range, and the rest of the distribution in respective order, except for points that were determined to be outliers using a method that is a function of the interquartile range, as in standard box plots.

CDE+ genes can enhance their fitness and in a subset of cases lead to selective outgrowth of KRAS mutant over WT cells in a mixed population.

**Cas9 expression in cancer cell lines selects for KRAS mutations.** Multiple studies have reported a higher editing efficiency of Cas9 in p53 mutated versus p53 WT cell lines[13–15,32]. We first asked if this may also extend to KRAS as an equally important CCD. Analyzing induced exogenous Cas9 activity in 1601 cancer cell lines from DepMap (1375 and 226 KRAS WT and mutant, respectively)[10], we find that, like p53, Cas9 activity is significantly higher in KRAS-mutant cells than in KRAS WT cells ($P = 2.9E−05$, Wilcoxon Rank-Sum test, Fig. 6d; see the "Methods" section). We repeated the above analysis modeling Cas9 activity vs. KRAS status adjusting for p53 status in a linear model, yielding concordant findings ($P = 2.43E−04$, Wilcoxon Rank-Sum test). Importantly, across all the 61 cancer driver genes we analyzed above from Vogelstein et al.[26], KRAS and p53 are the only ones showing such a significant difference in Cas9 activity between WT and mutant cells after FDR correction ($FDR = 9.1E−04$ for KRAS and $7.1E−06$ for p53, Wilcoxon Rank-Sum test, Fig. 6e). This further shows that in addition to p53, KRAS WT status can also hamper the efficiency of CRISPR-Cas9.

Based on the above findings and a recent report[15] of selection for p53 mutant due to DNA damage upon Cas9 expression (without sgRNA), we asked whether DNA damage induced by Cas9 alone can also lead to a mutation-selection of KRAS and/or other cancer drivers. To this end, we re-analyzed deep sequencing profiles from 42 Cas9-expressed vs. matched parental (i.e. without Cas9) cell lines (see the "Methods" section) from Enache et al.[15], and identified a total of 9 cases involving 5 unique KRAS mutations, occurring in 7 different cell lines with moderate to high Cas9 activity (Methods). Seven out of these 9 cases show increased mutant allele frequency after induced Cas9 expression (Wilcoxon signed-rank test $P = 0.027$, Fig. 6f). Four of the 5 KRAS mutations are missense mutations; the other mutation is an intronic mutation occurring 100 bp from the splicing site. This mutation is not present in the parental cell lines but emerges independently after Cas9 expression in four different cell lines.

While the results suggest that CRISPR-Cas9 may select for KRAS mutations, the functional role of these mutations needs to be interpreted with caution. Among the 61 cancer driver genes from Vogelstein et al.[26], KRAS is a top gene (ranked the second) along with p53 (ranked fourth) that shows significant mutant sub-clonal expansion (Fig. 6g). Notably, the top genes identified to be involved in mutant sub-clonal expansions have a significant overlap with our previously identified top CCD genes (Fisher's exact test $P = 0.04$). We note that when we repeated the above analysis excluding the intronic deletion, KRAS is ranked 16th out of 61 genes tested (Wilcoxon rank-sum $P < 0.28$).

**KRAS mutant cells downregulate the G2M checkpoint pathway in response to Cas9 induction.** To investigate the mechanism underlying the potential selective advantage of KRAS mutant vs. WT cells during CRISPR-KO, we analyzed gene expression data of 163 pairs of parental (without Cas9) and the corresponding Cas9-expressed cell lines (138 KRAS WT, 25 KRAS mutant)[15], and identified the pathways that are differentially regulated upon Cas9 expression between KRAS WT and mutant cells (i.e. up/down-regulated in KRAS mutant but inversely or non-significantly regulated in KRAS WT cells; Supplementary Fig. 9a, Supplementary Note 5 and Supplementary Fig. 10 for p53). A major differentially regulated pathway was the G2M checkpoint with the highest difference in the normalized enrichment score (Supplementary Fig. 9b), which is strongly downregulated (rank 2/50) in KRAS mutants but strongly upregulated in KRAS WT cells (4/50). This is a canonical pathway that serves to prevent the cells with genomic DNA damage from entering mitosis (M-phase) and thus its downregulation in KRAS mutant cells may provide them with a proliferative advantage[33]. Another top pathway, E2F Targets, which primarily regulates G1/S transition and DNA replication was also found to be downregulated in KRAS mutant cells but upregulated in KRAS WT cells upon Cas9 expression (Supplementary Fig. 9b). Thus, Cas9-induction may similarly underlie the selective advantage of KRAS mutant cells by selectively activating cell cycle checkpoint pathways in response to DNA damage.

## Discussion

In this study, we systematically investigated the possibility of selection of pre-existing cancer driver mutations during CRISPR-Cas9 gene editing. First, we confirmed and extended upon previous findings that selection[13–15] of pre-existing *p53* mutations by CRISPR-Cas9 can happen, showing it in a large set of transformed and non-transformed cell lines. We identified the specific CDE+ genes whose CRISPR-KO is likely to mediate such selection, and further tested and validated some of these predictions in new screens and competitive assays that we have performed. After studying and validating our integrated computational and experimental pipeline in the known case of *p53*, we turned to applying it to study a collection of major cancer driver genes and discovered that *KRAS* is another major cancer driver gene whose pre-existing mutants have a selective advantage during CRISPR-Cas9 gene editing. We demonstrated the selective advantage of *KRAS* mutant cells performing a CRISPR-KO/CRISPRi screen in isogenic cells with pooled CDE+ gene-targeting sgRNAs, and further in competition assays during the CRISPR-KO of top predicted *KRAS* CDE+ genes. We also observed that *KRAS* WT cells have lower Cas9 activity and thus a lower editing efficiency, similar to that observed for *p53*, which may limit CRISPR-mediated gene-editing in such cells[25]. Analyzing recently published *KRAS* screens, we also find a subclonal expansion of *KRAS* mutant cancer cells following Cas9 expression. Finally, our study also shows that the introduction of the Cas9 protein downregulates the G2M checkpoint and E2F targets in *KRAS* mutant, but not *KRAS* WT cells, which may confer a selective advantage to *KRAS*-mutant cells.

Multiple factors can contribute to the identity of CDE+ genes, including involvement in DNA repair and cell cycle pathways, being located in chromosomal fragile sites or HAC regions, supporting that their CRISPR-KO can lead to augmented DNA damage. We find that these factors can together account for up to 15% of our CDE+ genes. We also observed that a gene-targeted by highly off-target guides can also lead to high DNA damage, which reassuringly only accounts for up to 10% of the CDE+ genes (details in Supplementary Note 6). Taken together, these three putative mechanisms can explain about 25% of the CDE+ genes we have identified, however, the mechanisms underlying the rest are yet open to further studies.

Overall, our results point to a need for accounting for CDE effects in the analysis of dependencies in CRISPR screens. More importantly, our studies point to the need for careful selection of sgRNAs for therapeutic genome editing and recommend cautionary monitoring of *KRAS* status in addition to that of *p53* during therapies utilizing CRISPR-Cas9. Lastly, in the publicly available CRISPR-Cas9 screens that we have analyzed, the current small numbers of cell lines with mutations in other cancer drivers, such as *VHL*, limits our ability to reliably determine whether these cancer genes could also be selected during CRISPR-Cas9 genome editing. The investigation of the latter thus awaits specifically designed screens in designated isogenic cell lines.

## Methods

**CRISPR and shRNA essentiality screen data**. We obtained CRISPR-Cas9 essentiality screen (or dependency profile) data in 436 cell lines from Meyers et al.[10] for 16,368 genes, whose expression, CNV, and mutation data are available via the CCLE portal[34]. We obtained the shRNA essentiality screen data in 501 cell lines from the DepMap portal[35] for 16,165 genes, whose expression, CNV, and mutation data are available publicly via the CCLE portal[34]. The 248 cell lines and 14,718 genes that appear in both datasets were used in this analysis (Supplementary Data 1). For mutation data, only non-synonymous mutations were considered. Synonymous (silent) mutations were removed from the pre-processed MAF files downloaded from the CCLE portal[34].

**Identifying CRISPR specific differentially essential genes of a potential CRISPR-selected cancer driver**. For a given CCD (e.g. *p53* or *KRAS*), we checked which gene's essentiality (viability after knockout) is significantly associated with the mutational status of the CCD using a Wilcoxon rank-sum test in the CRISPR and shRNA datasets, respectively (FDR < 0.1). *CRISPR-specific differentially essential positive* (CDE+) genes are those whose CRISPR-KO is significantly more viable when the CCD is mutated while their shRNA silencing is not, whereas analogously CDE− genes are those whose CRISPR-KO is significantly more viable when the CCD is WT while their shRNA silencing is not. We filtered out any candidate CDE genes whose copy number was also significantly associated with the given mutation to control for potentially spurious associations coming from copy number (we removed genes showing significant association (FDR < 0.1))—the exact procedure used is described below in the section titled "Identifying potential CRISPR-selected cancer drivers").

**Identifying CDEs considering the functional impact of mutations**. Out of a total of 248 cell lines that we analyzed, 173 cell lines (69.7%) have *p53* non-synonymous mutations. In addition to identifying CDEs by considering all non-synonymous mutations, we additionally employed a more conservative approach where we aimed to consider only *p53* LOF mutations in the CDE identification process. To this end, we considered a mutation to be LOF if it was classified as non-sense, indel, frameshift, or among the 4 most frequent non-functional hotspot mutations (R248Q, R273H, R248W and R175H within the DNA-binding domain, determined as pathogenic by COSMIC[36]). Using this definition we obtained new mutation profiles for *p53* and identified CDE genes via the same method described in the section titled "Identifying CRISPR specific differentially essential genes of a potential CRISPR-selected cancer driver". We repeated a similar process with the top three known gain-of-function hotspot mutation variants of *KRAS*.

**Identifying potential CRISPR-selected cancer drivers of CRISPR-KO**. To identify additional CCD genes like *p53*, we considered 121 cancer driver genes identified by Vogelstein et al.[26], whose nonsynonymous mutation is observed in at least 10 cell lines ($N = 61$). We determined whether each of these genes is a CCD as follows: for each of the 61 candidate genes, we tested the association between the essentiality of each of genes in the genome (reflected by post-KO cell viability) with the mutational status of the candidate CCD gene using a Wilcoxon rank-sum test. We then counted the number of genes, whose essentiality is: (i) significantly positively associated with the candidate CCD mutational status (FDR-corrected $P$-value < 0.1, median essentiality of WT > mutant of the cancer gene), (ii) significantly negatively associated with the candidate CCD mutational status (FDR-corrected $P$-value < 0.1, median essentiality of WT < mutant of the cancer gene), and (iii) not associated (FDR-corrected $P$-value > 0.1) with the candidate CCD mutation status; we performed this computation separately for the CRISPR and the shRNA screens, respectively. This computation results in a 3-by-2 contingency table for each candidate CCD gene. We then checked whether the distribution of the above three counts in the CRISPR dataset significantly deviates from that in the shRNA dataset via a Fisher's exact test on the contingency table. If each of the values in the contingency table was >30, we used the chi-squared approximation of Fisher's exact test. We further filtered out any candidate CDE genes whose copy number was also significantly associated with the given mutation to control for potentially spurious associations coming from copy number (we removed genes showing significant association (FDR < 0.1)). We performed this procedure for all 61 candidate genes one by one and selected those with FDR corrected Fisher's exact test < 0.1. We further filtered out the candidate CCD whose mutation profile is correlated with *p53* mutation profile via a pairwise Fisher test of independence (FDR < 0.1). We finally report the CCD genes that have a substantial number of CDE+ genes ($N > 300$).

**Pathway enrichment analysis of CDE+/CDE− genes**. We analyzed the CDE +/CDE− genes of each of the CCDs for their pathway enrichment with annotations from the Reactome database[37] in two different ways. First, we tested for significant overlap between our CDE genes with each of the pathways with hypergeometric tests (FDR < 0.1). Second, we ranked all the genes in the CRISPR-KO screen by the differences in their median post-KO cell viability values in mutant vs. WT cells, and the standard GSEA method[38] was employed to test whether the genes of each Reactome pathway have significantly higher or lower ranks vs. the rest of the genes (FDR < 0.1). We repeated the GSEA analysis with the genes ranked by differential post-KD cell viability in the shRNA screen and only reported significant pathways specific to CRISPR but not shRNA screens. We confirmed that for *p53*, the GSEA method was able to recover the top significant pathways identified by the hypergeometric test (e.g. those in Fig. 1d), although extra significant pathways were identified (Supplementary Data 2). For *p53* and *KRAS* CDE− genes respectively, the enriched pathways were clustered based on the Jaccard index and the number of overlapping genes with Enrichment Map[39], and the largest clusters were visualized as network diagrams with Cytoscape[40]. To study the potential enrichment of CDE genes in CFSs, we obtained chromosomal band locations of CFS[17] and defined the CFS gene set as the set of all

genes located within these chromosomal bands (obtained from Biomart[41]). We tested for significant overlap between our CDE genes and the CFS gene set with a hypergeometric test, and also confirmed the lack of significant overlap with the corresponding shRNA-DE genesets. Similarly, for the common HAC regions, we obtained a list of these regions defined by a consensus of DNAseI and FAIRE across seven different cancer cell lines from a previous study[42]. Next, we identified sgRNAs that are expected to target such HAC regions (see the "Calculating off-target scores" section) and ranked genes based on the number of targeting such sgRNAs. Taking the top genes equal to the number of *p53* CDE+ genes, we computed the enrichment for *p53* CDE+ genes via a hypergeometric test.

**Testing the clinical relevance of copy number alterations of the p53 or KRAS CDE genes.** We tested the hypothesis that copy number alterations in CDE+ genes (as a possible surrogate for the number of DSBs in these genes) can reduce the fitness of the CCD (*p53* or *KRAS*) WT tumors with patient data. The cancer genome atlas (TCGA)[31] data of somatic copy number alteration (SCNA) and patient survival of 7547 samples in 26 tumor types were downloaded from the UCSC Xena browser (https://xenabrowser.net/). In these tumor types, *p53* is mutated in more than 5% of the samples. For each sample, the copy number alterations (GI) of a given set of genes, which quantifies the relative amplification or deletion of genes in a tumor based on SCNA was computed as follows[43]:

$$GI = \frac{1}{N} \sum_i I(s_i > 1) \quad (1)$$

where $s_i$ is the absolute log ratio of SCNA of gene $i$ in a sample relative to normal control, and $I()$ is the indicator function. Wilcoxon rank-sum test was then used to test whether the GI of CDE+ geneset is significantly lower than that of control non-CDE genes in CCD-WT but not in CCD-mutant tumors. Further, we tested if higher absolute levels of SCNA of the CDE+ genes are associated with increased rate of CCD (*p53* or *KRAS*) mutation accumulation with cancer stage, as this would further testify that such amplification/deletion events in the CDE+ genes can drive the selection for CCD mutants. To this end, the following logistic regression model was used to identify the genes whose high absolute SCNA computed as above is associated with a higher rate of CCD mutation accumulation with cancer stage, while controlling for cancer type and overall mutation load:

$$\text{logit}(P(\text{CCD})) = \beta_0 + \sum_k \beta_{\text{caner\_type\_k}} \text{cancer\_type}_k + \beta_{\text{mutation\_load}} \text{mutation\_load}$$
$$+ \beta_{\text{GI}} \text{GI}_i + \beta_{\text{stage}} \text{stage} + \beta_{\text{interact}} \text{GI}_i \cdot \text{stage}$$
$$(2)$$

where CCD denotes the binary CCD mutational status of the patient, $\text{logit}(P(\text{CCD}))$ is the logit function of the probability of the CCD being mutant; cancer_type$_k$ is the dummy variable for the category of the $k$th cancer type; GI$_i$ denotes the absolute value of SCNA levels of the given gene $i$ as computed above; GI$_i$ * stage is the interaction term between the GI of gene $i$ and cancer stage, that latter is made into a binary variable whose value is 0 for early stages (I and II) and 1 for late stages (III and IV). We tested the enrichment of CDE+ genes among the genes whose high absolute SCNA levels are significantly associated with a higher rate of CCD mutation accumulation with cancer stage (i.e. genes with significantly positive $\beta_{\text{interact}}$ coefficients in the above model) using a hypergeometric test.

**Constructs and stable cell lines.** MOLM13 cells were obtained from DSMZ (Cat. ACC-554) and maintained in RPMI-1640 medium (Life Technologies, Carlsbad, CA) supplemented with 10% v/v heat-inactivated fetal bovine serum (Sigma-Aldrich, Saint Louis, MI), 2 mM L-Glutamine (LifeTechnologies), and 100 U/mL penicillin/strepto-mycin (LifeTechnologies). *p53* R248Q was PCR amplified from a bacterial expression plasmid (kind gift of Dr. Shannon Lauberth, UCSD) and *KRAS*G12D the pBabe-*KRAS*G12D plasmid (Addgene plasmid 58902, from Dr. Channing Der) using the Kappa Hi-fidelity DNA polymerase (Kappa Biosystems). These PCR amplicons were separately cloned into the MSCV-IRES-tdTomato (pMIT) vector (a kind gift from Dr. Hasan Jumaa, Ulm) using Gibson Assembly. We first generated high-efficiency Cas9-editing MOLM13 leukemia cells by transducing these cells with the pLenti-Cas9-blasticidin construct (Addgene plasmid 52962—from Dr. Feng Zhang) and selecting stable clones using flow-sorting. Clones were then tested for editing efficiency by performing TIDE analysis[44]. These MOLM13-Cas9 cells were then transduced ret-rovirally with the pMIT-*p53*R248Q or pMIT-*KRAS*G12D mutants and sorted for tdTomato using flow-cytometry (LSR Fortessa, BD Biosciences) to generate isogenic mutant MOLM13-Cas9 cell lines. Immortalized hTERT RPE1 cells were obtained from ATCC® (Cat. CRL-4000™) and maintained in DMEM-F12 medium (Life Technologies, Carlsbad, CA) supplemented with 10% v/v heat-inactivated fetal bovine serum (Sigma-Aldrich, Saint Louis, MI), 2 mM L-glutamine (LifeTechnologies) and 100 U/mL penicillin/streptomycin (LifeTechnologies).

**Generation of pooled sgRNA libraries.** For pooled library cloning, 10 sgRNAs per gene were designed using the gene perturbation platform (https://portals.broad institute.org/gpp/public/analysis-tools/sgrna-design, Supplementary Data 9) Genetic Perturbation Platform. Guides targeting *p53* CDE+ and CDE− genes were synthesized as pools using array-based synthesis and cloned in the Lentiguide puro vector (Addgene plasmid 52963—a kind gift from Dr. Feng Zhang) using Golden Gate Assembly. In

each assay, we have used ~240 unique non-targeting sgRNAs and 49 not expressing non-essential genes. A similar approach was used for the *KRAS* CDE libraries.

**Pooled sgRNA library screen.** 30 million MOLM13-Cas9 cells or their isogenic MOLM13-*p53* or *KRAS* mutant counterparts were transduced with the pooled CDE library virus in RPMI medium supplemented with 10% fetal bovine serum, antibiotics, and 8 μg/ml polybrene. The medium was changed 24 h after transduction to remove the polybrene and cells were plated in a fresh culture medium. 48 h after transduction, puromycin was added at a concentration of 1 μg/ml to select for cells transduced with the sgRNA library. Puromycin was removed after 72 h and then cells were cultured for up to 30 days. 7 days after transduction, approximately 4 million cells were collected, and genomic DNA was prepared for the time zero (T0) measurement and also from time 30 (T30). Genomic DNA from these cells was used for PCR amplification of sgRNAs and sequenced using a MiSeq system (Illumina). Fold depletion or enrichment of sgRNAs from the NGS data was calculated using PinAplPy software[45].

**CDE+/− genes identified in isogenic experiments.** From the read counts per million for each sgRNA at Day 0 and Day 30 from the above-pooled CRISPR screens across two replicates, we removed all the sgRNAs with read count < 20 at Day 0. We calculated an average fold change (FC) of reads from Day 0 to Day 30. For each sgRNA, we calculated this FC-rank difference in *p53* WT vs mutant in both CRISPR-KO and CRISPRi screens. For consistent comparison with AVANA, we only considered sgRNAs used in both libraries. The top and bottom genes are *differentially essential (DE)* from each screen. Taking the top-ranked genes based on the difference of this score in two screens, we identify the CDE+ and CDE− genes.

**CRISPR competition experiments.** sgRNAs were cloned using standard cloning protocols and lentiviral supernatants were made from these sgRNAs in the 96-well arrayed format. 100,000 MOLM13 cells or tdTomato-positive isogenic mutants were plated in a 96-well plate and transduced with the sgRNA viral supernatants by spinfection with the polybrene-supplemented medium. After selection of sgRNA-transduced cells with puromycin for 48 h, sgRNA transduced MOLM13 cells or mutants were mixed in a ratio of 95:5, respectively. Cells were maintained in culture in 96-well plate format and assayed for proliferation every 4 days. A sample >10% of the culture volume was stained with SytoxBlue (1:1000) in PBS and monitored for the percentage of *p53* WT or *p53* mutant cells progressively up to 25 days using high-throughput flow-cytometry[22]

**Quality control of publicly mined genetic screens used in the study.** We first obtained gold-standard essential and non-essential geneset from Hart et al.[46]. To test the quality of each genetic screen we computed an area under the precision-recall curve (AUPRC) using the average logFC across replicates and cell lines. In this study, we only considered the genetic screens with an AUROC > 0.6 (random model AUPRC = 0.5). We also employed this method to test the quality of our in-house generated genetic screens.

**CRISPR-Cas9 RNP transfection experiments.** We generated sgRNAs by in vitro transcription using the HiScribe™ T7 Quick High Yield RNA Synthesis Kit (New England Biolabs, Beverley, MA) and performed the RNP complex formation using TrueCut Cas9 Protein v2 (ThermoFisher Scientific, Waltham, MA) according to published protocols[47]. MOLM13 cells without Cas9 and expressing pMIT-*p53*R248Q or pMIT-*KRAS* were generated as described in the *Constructs and stable cell lines* section. 1M cells were transfected with NFDUFB6 sgRNA or NTC sgRNA in triplicates with 1 mg of Cas9 and 1 mg of RNA in 10 ml of Buffer R using the Neon™ transfection system (ThermoFisher Scientific; 1500 V, 20 ms, single pulse). Cells were maintained in culture for 48 h before harvest for imaging, dye-dilution, and editing estimation assays. For *NDUFB6* and NTC editing estimation, we used the Synthego Performance Analysis ICE tool according to the instructions, using un-transfected parental MOLM13 samples as controls and samples from 48 h post-transfection as the Day 0 initial timepoint and Day 10 as a final time point, in triplicates. For RPE1 experiments, mutant cells with *p53*R248Q or pMIT-*KRAS* similarly and transfections were performed using Lipofectamine™ CRISPRMAX™ Cas9 Transfection Reagent (ThermoFisher Scientific) according to the manu-facturer's instructions for 12-well plate format, in triplicates. Cell harvesting time-points were similar to those of MOLM-13.

**Dye-dilution experiments.** We used the CellTrace™ Violet Cell Proliferation Kit (ThermoFisher Scientific) to stain MOLM13-WT and MOLM13-p53 mutant cells transfected with Cas9 RNP complexed with NTC or NDUFB6 RNA, according to the manufacturer's instructions. Cells were maintained in culture in the dark and assayed by flow cytometry using the LSR Fortessa every 2 days for 14 days. FCS files were analyzed using FlowJo software.

**Analysis of γ-H2AX foci in MOLM13-Cas9 and MOLM13-p53 mutant cells.** MOLM13-WT and MOlM13-p53 mutant cells were left untreated or treated with 1 μM doxorubicin for 2 h at 37 °C 5% $CO_2$, which served as negative and positive controls for DNA damage mediated γ-H2AX foci formation, respectively. MOLM13-WT and MOLM13-p53 mutant cells transfected with Cas9 RNP

complexed with NTC or NDUFB6, and negative control cells were pelleted at 400×*g* for 5 min at 4 °C, washed two times in PBS, and fixed in 4% paraformaldehyde in PBS for overnight at 4 °C. The cells were washed two times in PBS and permeabilized in 0.25% triton X-100 in PBS for 5 min at room temperature. Following two washes with PBS, the cells were incubated in blocking buffer (3% BSA in PBS) for 30 min at room temperature and subsequently incubated with APC conjugated H2AX phospho (Serine 139) antibody (BioLegend; Cat #613415) at an antibody dilution of 1:200 in blocking buffer for overnight at 4 °C in dark. Cells were washed two times with PBS and resuspended in 150 µl PBS. Cell suspensions were spotted on poly-lysine-coated glass slides using cytospin (Cytospin 4; Thermo Scientific) centrifugation at 72×*g* for 4 min. Coverslips were mounted onto the slides using ProLong Gold antifade reagent with DAPI (Invitrogen) and cured overnight at room temperature in dark. Slides were imaged in Nikon A1R HD confocal microscope. Sequential z-sections were imaged using a ×60 oil objective and maximum projection images were obtained using the Nikon NIS-Elements platform.

**Cas9 activity in cancer cell lines with KRAS (or another cancer driver) WT vs. mutant.** We downloaded the exogenous Cas9 activity of 1601 cancer cell lines from the DepMap portal and their *KRAS* mutation status considering only non-synonymous variants profiled using whole-exome sequencing (1375 and 226 *KRAS* WT and mutant, respectively)[16]. We tested whether the Cas9 activity is higher in *KRAS* mutant vs. *KRAS* WT cell lines using a one-sided Wilcoxon rank-sum test. We repeated this process for each cancer driver gene and used the FDR corrected significance to rank them in addition to the fold change of Cas9 expression.

**Subclonal expansion of KRAS mutant in parent vs. high Cas9-expressed cell lines.** We downloaded the deep targeted sequencing of cancer driver genes performed on 42 parental and matched Cas9-expressed cancer cell lines from Enache et al.[15]. In this analysis, we discarded the cell lines with <20% Cas9 activity and thus low DNA damage. We asked whether mutant allele frequency of a cancer driver (e.g. *KRAS*) significantly increased in Cas9-expressed cell lines compared to matched parental cell lines using Wilcoxon signed-rank test. In this analysis, we have considered both intronic and exonic variants provided by sequencing.

**Analysis of differentially expressed pathways in KRAS wildtype and mutant cells in response to Cas9 induction.** Gene expression profiles of 163 pairs of parental (without Cas9) and the corresponding Cas9-expressed cell lines (138 *KRAS* WT, 25 *KRAS* mutant) were obtained from Enache et al.[15]. Differential expression analysis between the Cas9-expressed cells and the parental cells was performed for the *KRAS* WT and mutant cells separately, and GSEA analysis[38] (genes ranked by logFC) was performed to identify the hallmark pathways from MSigDB[48]. We next identified pathways that are differentially regulated upon Cas9 expression between *KRAS* WT and mutant cells. These include the pathways that are up-regulated in the *KRAS* mutant cells but down-regulated or non-significantly altered in the *KRAS* WT cells and vice versa. The pathways are ranked by the difference of normalized enrichment score in WT vs. mutant cells. This analysis is performed using the *fgsea* R package[38].

**Reporting summary**. Further information on research design is available in the Nature Research Reporting Summary linked to this article.

## Data availability
The required data from in-house screens, in their raw and processed form, to reproduce each step of results and figures can be accessed here: Sanju Sinha (2021). A systematic genome-wide mapping of oncogenic mutation selection during CRISPR-Cas9 genome editing. https://doi.org/10.5281/zenodo.5479111. Processed expression, mutation, copy number, CRISPR-Cas9 and shRNA pooled genetic screen data were derived from DepMap v19Q3 and can be found here (https://depmap.org/portal/)[10]. Copy number and mutation profile of all patient tumors available in TCGA were retrieved from the firehose pipeline (https://gdac.broadinstitute.org/)[31]. Functional and positional genesets were derived from MSigDB (https://www.gsea-msigdb.org/gsea/msigdb/)[48]. Cas9 activity in 1601 cell lines from DepMap and deep sequencing profiles of these Cas9-expressed vs. matched cell lines were derived from Enache et al. (2020)[15] (Supplementary Data) and the raw data can be found at BioProject accession number PRJNA545458. Source data are provided with this paper.

## Code availability
We have provided the scripts to reproduce each step (numbered) of results and main text and supplementary figures in a GitHub repository which can be accessed here: https://doi.org/10.5281/zenodo.547858[49]. Source data are provided with this paper.

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

## Acknowledgements

We acknowledge and thank the National Cancer Institute for providing financial and infrastructural support. We thank Curtis Harris, Andre Nussenzweig, Sridhar Hannenhalli and the members of Cancer Data Science Lab for insightful feedback. This research was supported in part by the Intramural Research Program of the National Institutes of Health, NCI. S.S. and K.C. are supported by the NCI–UMD Partnership for Integrative Cancer Research Program. A.J.D. would like to acknowledge the support of the National Cancer Institute of the National Institutes of Health under Award Number P30 CA030199, the Rally Foundation for Childhood Cancer Research, and Luke Tatsu Johnson Foundation under Award Number 19YIN45, an Emerging Scientist Award from the Children's Cancer Research Fund, and the V Foundation for Cancer Research (TVF) under Award Number DVP2019-015.

## Author contributions

E.R., A.J.D., and J.S.L. supervised the study. S.S., K.C., K.B., E.R., A.J.D., J.S.L. conceived and designed the study. S.S., K.B., and K.C. developed the methodology. K.B., S.S., K.C., P.J., A.D., A.J.D. designed and performed experiments. S.S., K.B., K.C., M.D.L., P.J., A.D., D.M.W., B.M.R., J.L., Z.A.R. analyzed and interpreted the data. S.S., K.B., K.C., M.D.L., P.J., A.D., D.M.W., B.M.R., J.L., Z.A.R., J.S.L., A.J.D., E.R. wrote, reviewed and revised the manuscript.

## Funding

## Competing interests

The authors declare no competing interests.
