## [Peer Review File · Nature Communications]

Reviewers' Comments:

Reviewer #1:

Remarks to the Author:

In this manuscript, Sinha et al. demonstrate that CRISPR-Cas9 editing can select for the inactivation of genes beyond TP53. First, the authors analyzed CRISPR and shRNA screens in TP53-mutant and TP53-WT cell lines, and identified genes that are more essential in the TP53-WT cell lines only in the CRISPR screens (termed CDE+ genes). Next, these CDE+ genes were confirmed in CRISPR-KO vs. CRISPRi screens of isogenic TP53-WT/TP53-mutant cell lines. A flow cytometry-based competition assay confirmed the selection against TP53-WT cells upon knockout of approximately half of the predicted genes. Next, the authors applied a similar analysis to additional genes and identified KRAS as another gene that is potentially selected against during CRISPR-Cas9 gene editing. Similar to TP53, CDE+ genes were identified and validated for KRAS, and a competition assay confirmed selection for KRAS mutants following CRISPR-KO of such genes.

Overall, this is a comprehensive study of an important topic. The p53-related findings are not so novel, as multiple studies from recent years reported p53 activation and selection for p53-inactivating mutations in CRISPR-Cas9 experiments. The extension of these findings to additional genes, and in particular to KRAS, is more novel and is of importance.

Several issues require further analysis or clarifications:

- 1) The skewness in the numbers of differentially essential genes from CRISPR-Cas9 vs. shRNA screens is used as a measure of CRISPR selection, as "the mutants of these genes may be selected for during CRISPR-KO". This may be true, but there are other potential explanations for these differences. These should be discussed in the main text (and not just in Supplementary Note 1).
- 2) The Cas9 activity analysis (Fig. 6e) suggests that similar to p53, KRAS WT status constitutes a barrier to CRISPR-Cas9 editing. This point would be strengthened considerably if the authors showed it in a functional competition assay – would KRAS-mut cells expand following Cas9 induction in a mixed population?
- 3) In the analysis of mutation allele frequencies before and after Cas9 expression (without sgRNA), one of the mutations is an intronic mutation that arose independently in four different cell lines. Can this analysis be repeated including only mutations that are expected to be functional (i.e., excluding intronic mutations)? Would KRAS still come up as a "hit" in such analysis? If not, can the authors provide any evidence for the functional relevance of this mutation – how can we know that this is not just a sequencing artifact?
- 4) Supplementary Note 1: "To exclude the possibility that this effect is due to a difference in the phenotypes resulting from mRNA transcript knockdown by shRNA as opposed to gene knockout due to CRISPR, we repeated the analysis for a subset of genes that were not expressed in any of the cell lines (i.e. read count being 0). Still, we observed lower viabilities of p53 WT vs mutant cells specifically in CRISPR screens (Wilcoxon Rank Sum $P < 0.001$; for KRAS $P < 4E-09$)." I am not sure that I understand this control – why would non-expressed genes be essential in any of the cell lines?
- 5) Together with TP53 and KRAS, VHL is the third gene that seems to be an outlier in the CDE+ skewness analysis shown in Fig. 5a. Can the authors elaborate on it? They briefly acknowledge its existence in the last paragraph of the Discussion, but it seems to merit further elaboration (and analyses).

Reviewer #2:

Remarks to the Author:

The authors present results from a systematic effort aiming at investigating the possibility that additional cancer driver mutations (beside the established ones in TP53) are positively selected upon CRISPR-cas9 genome editing.

Toward this goal the authors have designed a bioinformatics strategy and retrospectively analysed public available data from functional genetics screens (employing CRISPR-cas9 and shRNA libraries) of a panel of genomically characterised immortalised human cancer cell lines, from the Cancer Dependency Map portal.

After confirming that in such data the previously reported selection of pre-existing TP53 mutations is observed, the authors have investigated whether this happens invariantly across cell models and generally upon the knock out of any gene and/or genomic regions.

Briefly, the authors identify genes showing a significant differential essentiality, according to the analysed CRISPR screens' dataset but not the shRNA screens' dataset, when contrasting TP53 mutant versus wild-type cell line populations.

They report that TP53 mutations tend to be more selected when targeting chromosomal bands containing common fragile sites, highly accessible chromatin regions and genes involved in DNA repair pathways and damage response.

Then they go further experimentally verifying that (i) via a CRISPR-cas9 and a CRISPRi screen, these findings hold true also in p53 isogenic cell lines (WT and R248mutant) and are specific to the CRISPR-cas9 system; (ii) systematic knockout of genes that are more essential in TP53wt cell lines positively selects TP53 mutations via a competition assay.

Furthermore the authors propose and use a strategy to harvest other cancer driver mutations that are putatively positively selected upon CRISPR-cas9 gene editing.

Towards this goal, the authors start from a set of prior known cancer drivers and for each of them they apply the same computational pipeline used for TP53, i.e. they identify genes that are differentially essential when contrasting driver mutant vs wt when using the CRISPR but not in the shRNAs screens. Then they rank the cancer drivers based on this tendency finding a handful of significant hits, with at the top KRAS, TP53 and VHL. Finally, they verify experimentally and similarly to TP53 that also mutations in KRAS are positively selected in CRISPR-cas9 experiments.

This is a great and timely piece of work, reporting findings that will be of broad interests to the functional genomics community and might have an impact on how results from CRISPR-cas9 screens are analysed and interpreted.

The computational pipeline is well thought and the experimental outcomes supports the claims for TP53 and KRAS.

The subsequent harvesting of novel driver mutations positively selected by the CRISPR-cas9 system is a bit less convincing.

A number of not considered confounders might have biased this analysis.

1. First of all, how the authors can be sure that there are no other functional links and true synthetic lethality between the considered drivers and the corresponding differentially essential genes? I understand that they focus on hits that are specific to the CRISPR-cas9 system (and not shRNA) but isn't this technology much more powerful, and precise than shRNA? Some hits might be simply observed in the CRISPR-Cas9 screen only because of this. This should be investigated, discussed and a method to estimate functional links outside the DNA repair / Damage response machinery that are not accounted and filtered out because of the lower efficiency of shRNA vs CRISPR should be devised.

2. I understand that for the purpose of characterising which pathway/genes/regions select more TP53mutants when targeted with CRISPR, it makes sense to focus on individual genes but to gain numerical power and mask possible confounders wouldn't make sense, in the subsequent analysis (when harvesting other positively selected drivers) to focus on bulk phenotypic readers? For example an estimator of the general susceptibility to CRISPR-cas9 targeting of the Xwt vs mut cell line (where X is the driver under consideration) might be the distance between the distribution of depletion fold-changes of prior known essential genes (for example ribosomal protein genes) and that of prior known non-essential genes (for example from the works published by Traver Hart).

3. Other confounders might be related to heterogeneous sgRNA efficiency and/or time dependent.

I do believe that to dilute CRISPR library specific biases the authors should intersect the differentially expressed genes found in the AVANA data (screened at 21days) with that coming from the Sanger screen (performed at 14days), and then they should remove the hits from the shRNA ACHILLES screen. Alternatively they could reperform their analysis on the recently published joint Sanger/Broad dependency dataset, where these study specific differences are robustly batch corrected (PMID: 33712601)

4. instead of seeking for novel positively selected mutations only in established cancer drivers, would't make sense to look also at immediate TP53 interactors that are somatically mutated in the considered panel of cell lines, for example TP53BP1 ?

5. The authors should make sure to use the largest and more recent releases of the DepMap data. Alternatively they could use the joint dataset which I mention in point 3.

6. Some of the supplementary tables contain dates instead of gene symbols (I suspect) because of the usual conversion problem in Microsoft Excel. This is very annoying. I would require to the authors to double check all the tables and to correct these errors.

Finally, while mentioning the problem related to potential gene-independent responses to CRISPR-cas9 targeting, the authors might add that other studies have reported that this type of bias is not always linearly correlated with gene copy numbers and that it is specific to tandem duplications, citing the following two studies, respectively, PMID: 30103702 and PMID: 30722791

We thank the editor and reviewers for their very helpful comments and have revised the manuscript accordingly. We believe that this has significantly further contributed to our study and the manuscript. Our response to each of the reviewer's comments (**bold format**) is stated below (**blue color text**). New information added in each section in the main manuscript, as well as supplemental sections, is also provided below (*blue color text in italics*).

Comments and responses:

Reviewer #1:

In this manuscript, Sinha et al. demonstrate that CRISPR-Cas9 editing can select for the inactivation of genes beyond TP53. First, the authors analyzed CRISPR and shRNA screens in TP53-mutant and TP53-WT cell lines, and identified genes that are more essential in the TP53-WT cell lines only in the CRISPR screens (termed CDE+ genes). Next, these CDE+ genes were confirmed in CRISPR-KO vs. CRISPRi screens of isogenic TP53-WT/TP53-mutant cell lines. A flow cytometry-based competition assay confirmed the selection against TP53-WT cells upon knockout of approximately half of the predicted genes. Next, the authors applied a similar analysis to additional genes and identified KRAS as another gene that is potentially selected against during CRISPR-Cas9 gene editing. Similar to TP53, CDE+ genes were identified and validated for KRAS, and a competition assay confirmed selection for KRAS mutants following CRISPR-KO of such genes. Overall, this is a comprehensive study of an important topic. The p53-related findings are not so novel, as multiple studies from recent years reported p53 activation and selection for p53-inactivating mutations in CRISPR-Cas9 experiments. The extension of these findings to additional genes, and in particular to KRAS, is more novel and is of importance.

We thank the reviewer for their encouraging words. We want to note that one of the main results of our study is the list of genes and pathways whose KO can lead to the emergence of cells with oncogenic mutations. Given how rapidly the field of CRISPR gene editing is moving into human patients, we believe that this knowledge is very important for careful consideration of clinical genome editing – clearly one of the most exciting advances in recent times.

1) The skewness in the numbers of differentially essential genes from CRISPR-Cas9 vs. shRNA screens is used as a measure of CRISPR selection, as “the mutants of these genes may be selected for during CRISPR-KO”. This may be true, but there are other potential explanations for these differences. These should be discussed in the main text (and not just in Supplementary Note 1).

We thank the reviewer for this important comment. In response, we added the following expanded section to the main text (page 5).

*“We next tested for a series of confounding factors that can potentially lead to this skewness (expanded details in **Supp. Note 1, 2, 6**). We repeated our analysis in the following modes to control for: 1. Gene copy number variations in cell lines, by correcting for gene copy number via a linear regression while testing whether a gene is differentially essential in WT vs mutant cell lines (**Figure S1**), 2. Functional effect of different p53 variants, by focusing on cell lines harboring known loss-of-function p53 mutations solely (Methods, N=78) vs WT (**Supp Notes 1a**), 3. Effect of partial vs complete silencing of gene expression in case shRNA-KD and CRISPR-Cas9 KO respectively, by using genes which are not expressed at all (**Supp Notes 1b**), 4. Effect of potential functional relationship eg. synthetic lethal or rescue interaction with p53, by using non-essential genes (**Supp Notes 1c**), 5. Effect of different reagents and sgRNA depletion time, by using an independent CRISPR-Cas9 screen (Sanger Screen) performed using in 326 cancer cell lines²⁰ (**Supp. Note 2, Figure S2**), 6. Off Target Effect of sgRNA, by computing and taking into account an off-target score of the sgRNA used in the*

screens (Supp Note. 6). We showed that our findings via the CDE identification process remain valid after controlling for this broad array of potentially confounding factors.”

2) The Cas9 activity analysis (Fig. 6e) suggests that similar to p53, KRAS WT status constitutes a barrier to CRISPR-Cas9 editing. This point would be strengthened considerable if the authors showed it in a functional competition assay – would KRAS-mut cells expand following Cas9 induction in a mixed population?

Thanks. We have previously analyzed a competition assay performed by Enache et al. that aims to do this. Here, deep sequencing profiles of 42 Cas9-expressed vs matched parental (i.e. without Cas9) cell lines (see Methods) were generated. These 42 cell lines are a mixed population of cells comprising both KRAS WT and KRAS Mutant, where the ratio of WT vs Mutant is measured pre- and post-Cas9 induction by targeted deep sequencing of KRAS allele. Analyzing this dataset, we observed a significant KRAS mutant sub-clonal expansion. Repeating this test for 61 cancer driver genes from Vogelstein et al. [27], KRAS is a top gene (ranked the second) that shows significant mutant sub-clonal expansion. Specifically, this identified a total of 9 cases involving 5 unique KRAS mutations, occurring in 7 different cell lines with moderate to high Cas9 activity (Methods). Seven out of these 9 cases show increased mutant allele frequency after induced Cas9 expression (Wilcoxon signed-rank test $P=0.027$). Text to the corresponding analysis is provided in the section “Cas9 expression in cancer cell lines selected for KRAS mutations” in the main text (Page 17).

3) In the analysis of mutation allele frequencies before and after Cas9 expression (without sgRNA), one of the mutations is an intronic mutation that arose independently in four different cell lines. Can this analysis be repeated including only mutations that are expected to be functional (i.e., excluding intronic mutations)? Would KRAS still come up as a “hit” in such

analysis? If not, can the authors provide any evidence for the functional relevance of this mutation – how can we know that this is not just a sequencing artifact?

In response to this question, we repeated the analysis excluding all the intronic mutations (deletions), KRAS is ranked at 16th out of 61 drivers tested (Wilcoxon rank-sum $P < 0.28$). We note that excluding intronic mutations decreases the statistical power of this test which may contribute to this (Total variants=13,451, Total variants after excluding intronic variants=5510). We thank the reviewer for this comment and agree that performing this specific analysis in this manner does not show that currently known functional impact KRAS mutations are selected for. It is unlikely that this arising intronic mutation is a sequencing artifact since the mutation (c.451-100delA) is not present in the parental cell lines but emerges independently after Cas9 expression in four different cell lines. Furthermore, this mutation has been described in Leiden Open Variation Database of KRAS mutations (<https://databases.lovd.nl/shared/variants/KRAS>) – testifying its previous reports. We note that, aside of this analysis, we have demonstrated the selection of mutant KRAS through additional four independent ways, as follows:

- i) Computational analysis of multiple genome-wide CRISPR and shRNA screens (**Figure 5A-C**)
- ii) Differential essentiality in CRISPRko but not CRISPRi screens in isogenic screens (**Figure 6A-B**)
- iii) A FACS based competition assay using an isogenic cell line (**Figure 6C**)
- iv) Demonstrating that KRAS mutant cells downregulate the G2M checkpoint pathway in response to Cas9 induced double stranded breaks. (**Figure S9**)

For completeness, we added the following text explaining this analysis in the main text, section “*Cas9 expression in cancer cell lines selects for KRAS mutations*” (page 18).

“We note that when we repeated the above analysis excluding intronic deletion, KRAS is ranked 16th out of 61 genes (Wilcoxon rank-sum $P < 0.28$).”

4) Supplementary Note 1: “To exclude the possibility that this effect is due to a difference in the phenotypes resulting from mRNA transcript knockdown by shRNA as opposed to gene knockout due to CRISPR, we repeated the analysis for a subset of genes that were not expressed in any of the cell lines (i.e. read count being 0). Still, we observed lower viabilities of p53 WT vs mutant cells specifically in CRISPR screens (Wilcoxon Rank Sum $P < 0.001$; for KRAS $P < 4E-09$).” I am not sure that I understand this control – why would non-expressed genes be essential in any of the cell lines?

Thanks for raising this important point, which deserves a clearer explanation: shRNA-based Knockdown (KD) could potentially result in a partial silencing of gene activity in difference from the complete silencing occurring with gene knockout (KO) via CRISPR-Cas9. This could confound the identification process of CDE genes, in the cases where a gene might be differentially essential only when completely removed. To control for this potentially confounding effect, we repeated our analysis focusing just on the genes that are not expressed at all, thus eliminating this possible confounding effect of partial vs complete gene silencing. We still observed that KO of these non-expressed genes significantly selects for mutant cells specifically in the CRISPR vs shRNA screens (Wilcoxon Rank Sum $P < 0.001$; for KRAS $P < 4E-09$). This can only be attributed to CRISPR-mediated DNA double stranded breaks instead of effects of gene perturbation since the genes were not expressed. To improve the clarity of the text, we provided this expanded description in the Supp Text, Page 3, as follows:

Supp Note 1b: “Considering that shRNA knockdown (KD) silences a gene at the mRNA level, this method could potentially result in a partial ablation of gene activity in contrast to the complete silencing of gene expression observed with their knockout (KO) via CRISPR-Cas9. This could potentially confound the identification process of CDE genes, in the cases where a gene might be

differentially essential only when completely removed. To control for this potentially confounding effect, we repeated our analysis focusing just on the genes that are not expressed at all, thus eliminating this possible confounding effect of partial vs complete gene silencing. We still observed that KO of these non-expressed genes significantly selects for mutant cells specifically in the CRISPR vs shRNA screens (Wilcoxon Rank Sum $P < 0.001$; for KRAS $P < 4E-09$). This can only be attributed to CRISPR-mediated DNA double stranded breaks instead of effects of gene perturbation, since the genes in this analysis were transcriptionally inactive.

5) Together with TP53 and KRAS, VHL is the third gene that seems to be an outlier in the CDE+ skewness analysis shown in Fig. 5a. Can the authors elaborate on it? They briefly acknowledge its existence in the last paragraph of the Discussion, but it seems to merit further elaboration (and analyses).

Thank you for this important observation. Following, we added the following paragraph and analysis to the main text, page 14:

*“Based on the number of CDE+ genes (**Figure 5a**), the third-ranked CCD is VHL, having a large number of CDE+ genes. Like p53 and KRAS, we controlled for a series of potential confounding factors (**Supp Notes 1, 2, 6**). We also note that the CDE+ genes of VHL were also enriched in chromosomal bands of CFSs (hypergeometric $P < 2.4e-2$). The known function of VHL as a positive regulator of p53 in DNA damage-induced cell cycle arrest or apoptosis¹¹ possibly accounts for its role as a CCD. However, in this study, we chose to carefully study one additional CCD and our top-ranked hit - KRAS and its potential mutant selection during CRISPR-Cas9 gene editing and thus will focus on that from here onwards.”*

Reviewer #2:

The authors present results from a systematic effort aiming at investigating the possibility that additional cancer driver mutations (beside the established ones in TP53) are positively selected upon CRISPR-cas9 genome editing. Toward this goal the authors have designed a bioinformatics strategy and retrospectively analysed public available data from functional genetics screens (employing CRISPR-cas9 and shRNA libraries) of a panel of genomically characterised immortalised human cancer cell lines, from the Cancer Dependency Map portal. After confirming that in such data the previously reported selection of pre-existing TP53 mutations is observed, the authors have investigated whether this happens invariantly across cell models and generally upon the knock out of any gene and/or genomic regions. Briefly, the authors identify genes showing a significant differential essentiality, according to the analysed CRISPR screens' dataset but not the shRNA screens' dataset, when contrasting TP53 mutant versus wild-type cell line populations. They report that TP53 mutations tend to be more selected when targeting chromosomal bands containing common fragile sites, highly accessible chromatin regions and genes involved in DNA repair pathways and damage response. Then they go further experimentally verifying that (i) via a CRISPR-cas9 and a CRISPRi screen, these findings hold true also in p53 isogenic cell lines (WT and R248mutant) and are specific to the CRISPR-cas9 system; (ii) systematic knockout of genes that are more essential in TP53wt cell lines positively selects TP53 mutations via a competition assay. Furthermore the authors propose and use a strategy to harvest other cancer driver mutations that are putatively positively selected upon CRISPR-cas9 gene editing. Towards this goal, the authors start from a set of prior known cancer drivers and for each of them they apply the same computational pipeline used for TP53, i.e. they identify genes that are differentially essential when contrasting driver mutant vs wt when using the CRISPR but not in the shRNAs screens. Then they rank the cancer drivers based on this tendency finding a handful of significant hits, with at the top KRAS, TP53 and VHL. Finally, they verify experimentally and similarly to TP53 that also mutations in KRAS are

positively selected in CRISPR-cas9 experiments. This is a great and timely piece of work, reporting findings that will be of broad interests to the functional genomics community and might have an impact on how results from CRISPR-cas9 screens are analysed and interpreted. The computational pipeline is well thought and the experimental outcomes supports the claims for TP53 and KRAS. The subsequent harvesting of novel driver mutations positively selected by the CRISPR-cas9 system is a bit less convincing. A number of not considered confounders might have biased this analysis.

We thank the reviewer for the encouraging comments and have addressed the concerns to the best of our abilities as noted below.

1. First of all, how the authors can be sure that there are no other functional links and true synthetic lethalties between the considered drivers and the corresponding differentially essential genes? I understand that they focus on hits that are specific to the CRISPR-cas9 system (and not shRNA) but isn't this technology much more powerful, and precise than shRNA? Some hits might be simply observed in the CRISPR-Cas9 screen only because of this. This should be investigated, discussed and a method to estimate functional links outside the DNA repair / Damage response machinery that are not accounted and filtered out because of the lower efficiency of shRNA vs CRISPR should be devised.

We thank the reviewer for this important comment. In our study, we tested for a series of confounding variables in our CDE identification process including the ones that the reviewer has pointed out. We did this by re-running our framework at various different modes that are described here to control for:

1. Gene copy number in cell line, by correcting for gene copy number via a linear regression while testing whether a gene is differentially essential in WT vs mutant cell lines (**Figure S1**),

2. Functional effect of different p53 variants, by focusing on cell lines harboring known loss-of-function p53 mutations solely (Methods, N=78) vs WT,

3. Effect of partial vs complete silencing of gene expression in case shRNA-KD and CRISPR-Cas9 KO respectively, by using genes which are not expressed at all,

4. Effect of potential functional relationship eg. synthetic lethal or rescue interaction with p53, by using non-essential genes,

5. Effect of different reagents and sgRNA depletion time, by using independent CRISPR-Cas9 screen (Sanger Screen) performed using in 326 cancer cell lines²⁰ (**Supp. Note 2, Figure S2**),

6. Off-Target Effects of sgRNA, by computing and taking into account an off-target score of the sgRNA used in the screens (Supplementary Note. 6).

We showed that our findings via the CDE identification process remain valid after controlling for this broad array of potentially confounding factors. However, this has not been previously carefully discussed in the main text comprehensively. Addressing this, we added the following response to the main text, page 5: *Section “CRISPR-Cas9 gene-knockouts selects for p53 mutations in a vast variety of transformed and non-transformed cell types”:*

*“We next tested for a series of confounding factors that can potentially lead to this skewness (expanded details in **Supp. Note 1, 2, 6**). We repeated our analysis in the following modes to control for: 1. Gene copy number variations in cell lines, by correcting for gene copy number via a linear regression while testing whether a gene is differentially essential in WT vs mutant cell lines (**Figure S1**), 2. Functional effect of different p53 variants, by focusing on cell lines harboring known loss-of-function p53 mutations solely (Methods, N=78) vs WT (**Supp Notes 1a**), 3. Effect of partial vs complete silencing of gene expression in case shRNA-KD and CRISPR-Cas9 KO respectively, by using genes which are not expressed at all (**Supp Notes 1b**), 4. Effect of potential functional relationship eg. synthetic lethal or rescue interaction with p53, by using non-essential genes (**Supp Notes 1c**), 5. Effect of different reagents and sgRNA depletion time, by using independent CRISPR-Cas9 screen (Sanger*

Screen) performed using in 326 cancer cell lines²⁰ (*Supp. Note 2, Figure S2*), 6. *Off Target Effect of sgRNA*, by computing and taking into account an off-target score of the sgRNA used in the screens (*Supp Note. 6*). We showed that our findings via the CDE identification process remain valid after controlling for this broad array of potentially confounding factors.”

2. I understand that for the purpose of characterising which pathway/genes/regions select more TP53mutants when targeted with CRISPR, it makes sense to focus on individual genes but to gain numerical power and mask possible confounders wouldn't make sense, in the subsequent analysis (when harvesting other positively selected drivers) to focus on bulk phenotypic readers? for example an estimator of the general susceptibility to CRISPR-cas9 targeting of the Xwt vs mut cell line (where X is the driver under consideration) might be the distance between the distribution of depletion fold-changes of prior known essential genes (for example ribosomal protein genes) and that of prior known non-essential genes (for example from the works published by Traver Hart).

We thank the reviewer for this interesting comment and suggestion. However, we note that the essential and non-essential gene sets the reviewer is referring to (Traver Hart et al. 2015, Cell) are computed via an shRNA screen solely, which may confound this analysis. However, still aiming to study and address your suggestion further as best as we could, we computed this susceptibility as the mean difference between genes essentiality in WT vs mutant cell lines for X driver gene (where X is the driver under consideration) in the CRISPR-Cas9 screen, vs this score in the shRNA screen. To overcome screen-specific variance, we centered and standardized the standard deviation of both datasets. To identify drivers whose selection is specific to CRISPR-KO, we looked for drivers whose susceptibility scores are very high in the CRISPR-Cas9 screen but low in shRNA-screen (**Figure R1**). In this analysis, we observed that the identity of the genes ranked high by susceptibility scores in the

CRISPR-Cas9 screen but low in shRNA-screen (rank difference) is dependent on the set of genes used: When essential genes are used to compute this score, RB1 and TP53 are the top two ranked genes and KRAS is ranked 23rd (**Figure R1A**). When non-essential genes are used, VHL and CDKN2A are the top two hits and TP53 and KRAS are ranked 47th and 40th (**Figure R1B**), respectively. We also do not observe an overall concordance between the two rank lists (Pearson Rho=-0.35). We reiterate that both the essential and non-essential gene sets were derived from shRNA screens and thus this analysis is likely to be quite confounded. We hence think it probably makes better sense not to include it in our report, but it could be added as supplementary material if the reviewer and the editor think that this is of interest.

Review Figure 1: Potential Mutant selection susceptibility computed using the difference in essentiality score between driver Mut vs WT cell lines in CRISPR and shRNA screens using essential and non-essential genes. The mean difference in essentiality scores between driver Mut vs WT cell lines is computed using shRNA screens (X-axis) and CRISPR-Cas9 screen (Y-axis) using (A) essential genes and (B) non-essential genes. This is computed for 61 driver genes identified by Vogelstein et al (each point) and labeled accordingly.

3. Other confounders might be related to heterogeneous sgRNA efficiency and/or time dependent. I do believe that to dilute CRISPR library specific biases the authors should intersect the differentially expressed genes found in the AVANA data (screened at 21days) with that coming from the Sanger screen (performed at 14days), and then they should remove the hits from the shRNA ACHILLES screen. Alternatively, they could reperform their analysis on the recently published joint Sanger/Broad dependency dataset, where these study specific differences are robustly batch corrected (PMID: 33712601)

Thanks. Indeed, compared to the DepMap, the Sanger Institute's screens include a higher number of sgRNAs per gene and a lower assay dropout length. We mined the shRNA data available for the subset of cell lines used in both these screens and repeated the identification of CDE genes and CCDs. First, we indeed observed a high number of p53 DE+ genes (N=752) compared to DE- (N=58), the majority of them being CRISPR-screen specific. In contrast, the shRNA screens again had a balanced number of DE+ and DE- genes (Chi-squared imbalance test $P < 1.4E-284$). We next observed that both CDE+ and CDE- genes identified from both the screens are significantly overlapping (hypergeometric $P < 2E-63$ and $< 8E-07$, respectively). Next, we repeated the process of CRISPR-selected cancer drivers (CCD) identification and obtained *KRAS* and *p53* as our two hits with similar and highly overlapping *KRAS* CDE+ and CDE- genesets (hypergeometric $P < 2E-108$ and $< 1.3E-19$, respectively). We briefly mention this in the main text, section "*CRISPR-Cas9 gene-knockouts selects for p53 mutations in a vast variety of transformed and non-transformed cell types*" (Page 5), which reads as follows:

*"Effect of different reagents and sgRNA depletion time, by using independent CRISPR-Cas9 screen (Sanger Screen) performed using in 326 cancer cell lines²⁰ (Supp. Note 2, Figure S2)"*

We also provided a comprehensive description of this analysis in **Supp Note 2** (Supp Text, Page 4) which reads as follows:

“During the process of completion of this manuscript, genome-wide CRISPR-Cas9 screens in 326 cancer cell lines were generated at the Sanger Institute through an independently designed experimental pipeline [1]. Compared to the DepMap, the Sanger Institute’s screens include a higher number of sgRNAs per gene and a lower assay dropout length. We mined the shRNA data available for the subset of cell lines used in these screens and repeated the identification of CDE genes and CCDs. First, we indeed identified a high number of DE+ genes (N=752) compared to DE- (N=58), the majority of them being CRISPR-screen specific. In contrast, the shRNA screens again had a balanced number of DE+ and DE- genes (Chi-squared imbalance test $P < 1.4E-284$). We next observed that both CDE+ and CDE- genes identified from both the screens were significantly overlapping (hypergeometric $P < 2E-63$ and $< 8E-07$, respectively). Next, we repeated the process of CRISPR-selected cancer drivers (CCD) identification and obtained KRAS and p53 as our two hits with similar and highly overlapping KRAS CDE+ and CDE- genesets (hypergeometric $P < 2E-108$ and $< 1.3E-19$, respectively).”

4. instead of seeking for novel positively selected mutations only in established cancer drivers, wouldn't it make sense to look also at immediate TP53 interactors that are somatically mutated in the considered panel of cell lines, for example TP53BP1?

We thank the referee for this insightful suggestion. In response to this, we analyzed the STRING database (<https://string-db.org/>), and identified the top 500 genes (Table S6) with the highest confidence interaction with p53. We first observed that most of these genes are not commonly mutated. We re-run the analysis with genes that are mutated in at least 1% of the cell lines panel used (N=96) and observed that only KRAS and VHL pass the Chi Sq. $P < 0.05$ having a large set of CDE+

genes (Review Figure 2). These are indeed the top two hits in our previous analysis and, in light of this analysis and a related comment by referee no. 1 (Comment #5), we have now added a paragraph briefly describing the VHL findings now (main text, page 14), which reads as follows:

“Based on the number of CDE+ genes (Figure 5a), the third-ranked CCD is VHL, having a large number of CDE+ genes. Like p53 and KRAS, we controlled for a series of potential confounding factors (Supp Notes 1, 2, 6). We also note that the CDE+ genes of VHL were also enriched in chromosomal bands of CFSs (hypergeometric $P < 2.4e-2$). The known function of VHL as a positive regulator of p53 in DNA damage-induced cell cycle arrest or apoptosis¹¹ possibly accounts for its role as a CCD. However, in this study, we chose to carefully study one additional CCD and our top-ranked hit - KRAS and its potential mutant selection during CRISPR-Cas9 gene editing and thus will focus on that from here onwards.”

Review Figure 2: A scatter plot showing the number of identified CDE+ genes (X-axis) and the negative log10-transformed P values of Fisher's exact test (Y-axis) testing for the imbalance in the

number of differentially essential genes in CRISPR and shRNA screens for the 96 interactors of p53 genes that are mutated in at least 1% of the cell lines panel used. KRAS and VHL are identified as the top two significant cancer genes with a higher number of CDE+ genes.

In closing, we should note that our ability to determine whether a cancer gene would be selected during CRISPR-Cas9 genome editing with high statistical confidence requires the gene to be mutated in at least a reasonable subset of cell lines used, and thus we cannot conclusively rule out the selection of the large subset of p53 interactors which are not mutated in at least a reasonable number of cell lines.

5. The authors should make sure to use the largest and more recent releases of the DepMap data. Alternatively, they could use the joint dataset which I mention in point 3.

Thanks. We mined the latest version of DepMap comprising 521 cell lines with both CRISPR-KO and shRNA-KD screens and repeated the identification of CDE genes and CCDs. Consistent with our previous findings, we observed a high number of DE+ genes (N=910) compared to DE- (N=31), where the majority of them being CRISPR-screen specific. In contrast, the shRNA screens again had a balanced number of DE+ and DE- genes (Chi-squared imbalance test $P < 3.8E-201$). We next observed that both CDE+ and CDE- genes identified from both the screens were significantly overlapping (J-index=0.78 and 0.24 for CDE+ and CDE-, respectively, hypergeometric $P < 2E-104$ and $< 3.2E-15$, respectively). Next, we repeated the process of CRISPR-selected cancer drivers (CCD) identification and obtained KRAS and p53 as our two top hits. Accordingly, we added text describing this analysis in **Supp Notes 13, “Re-running the analysis using the latest version of DepMap”**. The new text (Supp Text, Page 9-10) reads as follows:

“We mined the latest version of DepMap comprising 521 cell lines with both CRISPR-KO and shRNA-KD screens and repeated the identification of CDE genes and CCDs. Consistent with our previous

findings, we observed a high number of DE+ genes (N=910) compared to DE- (N=31), where the majority of them being CRISPR-screen specific. In contrast, the shRNA screens again had a balanced number of DE+ and DE- genes (Chi-squared imbalance test $P < 3.8E-201$). We next observed that both CDE+ and CDE- genes identified from both the screens were significantly overlapping (J-index=0.78 and 0.24 for CDE+ and CDE-, respectively, hypergeometric $P < 2E-104$ and $< 3.2E-15$, respectively). Next, we repeated the process of CRISPR-selected cancer drivers (CCD) identification and obtained KRAS and p53 as our two top hits.”

• **6. Some of the supplementary tables contain dates instead of gene symbols (I suspect) because of the usual conversion problem in Microsoft Excel. This is very annoying. I would require to the authors to double check all the tables and to correct these errors.**

Thanks. We sincerely apologize for any inconvenience this may have caused. We used a publicly available resource to identify gene names (N=30, <https://genomebiology.biomedcentral.com/articles/10.1186/s13059-016-1044-7>) whose HGNC symbol can be potentially converted into dates by Excel. Scanning across all our tables, we identify that Table S1B has 11 such instances. We fixed this by introducing quotes before and after the gene name.

Finally, while mentioning the problem related to potential gene-independent responses to CRISPR-cas9 targeting, the authors might add that other studies have reported that this type of bias is not always linearly correlated with gene copy numbers and that it is specific to tandem duplications, citing the following two studies, respectively, PMID: 30103702 and PMID: 30722791.

Thanks. Addressing this comment, we now added the following text in the Introduction citing the relevant papers (Main Text, Page 2):

“Other studies have demonstrated that double-stranded breaks (DSBs) induced during CRISPR-Cas9-based gene knockout (CRISPR-KO) can lead to DNA damage response, whose level can either be associated with the copy number of the targeted gene⁷⁻¹⁰ or in some cases to structural rearrangements in the region⁴⁷⁻⁴⁸.”

Reviewers' Comments:

Reviewer #1:

Remarks to the Author:

The authors have addressed my comments adequately. I have no further comments.

Reviewer #2:

Remarks to the Author:

The authors have put a lot of effort in addressing all mine and other reviewers' comments. As a result their manuscript (which already positively impressed me in its original form) is much improved.

I have only one final comment: with respect to my 2nd point, the authors decided not to include a new analysis arguing that the set of prior known essential genes I proposed (from Hart et al) is derived from shRNA studies only. While I totally agree with them on the unsuitability of this set, I still believe that the author might nevertheless include the analysis I suggested focusing on unbiased sets of prior known essential genes, for example derived from literature or the molecular signature database, i.e. ribosomal protein genes, genes involved in DNA replication, spliceosome etc.

Reviewer #2 (Remarks to the Author):

The authors have put a lot of effort in addressing all mine and other reviewers' comments. As a result their manuscript (which already positively impressed me in its original form) is much improved.

I have only one final comment: with respect to my 2nd point, the authors decided not to include a new analysis arguing that the set of prior known essential genes I proposed (from Hart et al) is derived from shRNA studies only. While I totally agree with them on the unsuitability of this set, I still believe that the author might nevertheless include the analysis I suggested focusing on unbiased sets of prior known essential genes, for example derived from literature or the molecular signature database, i.e. ribosomal protein genes, genes involved in DNA replication, spliceosome etc.

We repeated the suggested analysis using high confidence essential genes with known essential functions i.e. Ribosomal genes, and added the below text in the supplementary Notes 14 presenting results in Figure S13.

“14. Computing mutant selection susceptibility using known essential geneset

*We also computed the mutant selection susceptibility for each driver gene using the mean difference between genes essentiality of known essential genes (Ribosomal genes) in WT vs mutant cell lines in the CRISPR-Cas9 screen, vs this score in the shRNA screen. To overcome screen-specific variance, we centered and standardized the standard deviation of both datasets. To identify drivers whose selection is specific to CRISPR-KO, we looked for drivers whose susceptibility scores are very high in the CRISPR-Cas9 screen but low in shRNA-screen (**Figure S13**). Our top hits ranked by high susceptibility scores in the CRISPR-Cas9 screen but low in shRNA-screen (rank difference) are RB1, FOXL2 and CSF1R (Top three hits in respective order, **Figure S13**). Here, KRAS and p53 were ranked 7th and 11th, respectively. We note that the essential gene identity was derived from shRNA screens and thus this analysis may have confounding factors.”*